# INCREASING THE COST OF MODEL EXTRACTION WITH CALIBRATED PROOF OF WORK

**Adam Dziedzic**
University of Toronto and Vector Institute
adam.dziedzic@utoronto.ca

**Muhammad Ahmad Kaleem**
University of Toronto and Vector Institute
ahmad.kaleem@mail.utoronto.ca

**Yu Shen Lu**[*]
Stanford University
yushenlu@stanford.edu

**Nicolas Papernot**
University of Toronto and Vector Institute
nicolas.papernot@utoronto.ca

## ABSTRACT

In model extraction attacks, adversaries can steal a machine learning model exposed via a public API by repeatedly querying it and adjusting their own model based on obtained predictions. To prevent model stealing, existing defenses focus on detecting malicious queries, truncating, or distorting outputs, thus necessarily introducing a tradeoff between robustness and model utility for legitimate users. Instead, we propose to impede model extraction by requiring users to complete a proof-of-work before they can read the model's predictions. This deters attackers by greatly increasing (even up to 100x) the computational effort needed to leverage query access for model extraction. Since we calibrate the effort required to complete the proof-of-work to each query, this only introduces a slight overhead for regular users (up to 2x). To achieve this, our calibration applies tools from differential privacy to measure the information revealed by a query. Our method requires no modification of the victim model and can be applied by machine learning practitioners to guard their publicly exposed models against being easily stolen.

## 1 INTRODUCTION

Model extraction attacks (Tramèr et al., 2016; Jagielski et al., 2020; Zanella-Beguelin et al., 2021) are a threat to the confidentiality of machine learning (ML) models. They are also used as reconnaissance prior to mounting other attacks, for example, if an adversary wishes to disguise some spam message to get it past a target spam filter (Lowd & Meek, 2005), or generate adversarial examples (Biggio et al., 2013; Szegedy et al., 2014) using the extracted model (Papernot et al., 2017b). Furthermore, an adversary can extract a functionally similar model even without access to any real input training data (Krishna et al., 2020; Truong et al., 2021; Miura et al., 2021) while bypassing the long and expensive process of data procuring, cleaning, and preprocessing. This harms the interests of the model owner and infringes on their intellectual property.

Defenses against model extraction can be categorized as active, passive, or reactive. Current active defenses perturb the outputs to poison the training objective of an attacker (Orekondy et al., 2020). Passive defenses try to detect an attack (Juuti et al., 2019) or truncate outputs (Tramèr et al., 2016), but these methods lower the quality of results for legitimate users. The main reactive defenses against model extraction attacks are watermarking (Jia et al., 2020b), dataset inference (Maini et al., 2021), and proof of learning (Jia et al., 2021). However, reactive approaches address model extraction post hoc, *i.e.*, after the attack has been completed.

We design a pro-active defense that prevents model stealing *before* it succeeds. Specifically, we aim to increase the computational cost of model extraction without lowering the quality of model outputs. Our method is based on the concept of proof-of-work (PoW) and its main steps are presented as a block diagram in Figure 1. Our key proposal is to wrap model outputs in a PoW problem to force users to expand some compute before they can read the desired output. The standard PoW techniques,

---

[*]Work done while the author was at the University of Toronto and Vector Institute.

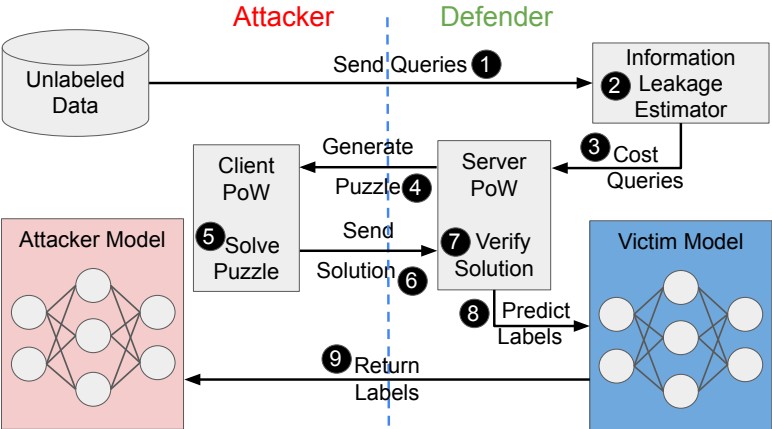

Figure 1: **Machine Learning API with calibrated proof-of-work.** Answering queries with PoW consists of the following steps: **1** An attacker sends queries drawn from an unlabeled dataset to a machine learning service. **2** The information leakage estimator computes the privacy cost of the queries. **3** The privacy cost of queries is sent to the proof-of-work (PoW) server, which adds the new cost to the total user's (attacker's) query cost. **4** The PoW server generates a puzzle, whose difficulty is based on the total cost incurred by the user. **5** The attacker solves the puzzle using its PoW Client and **6** sends the solution back to the server. **7** The PoW server verifies the solution and if it is correct, it notifies the victim model, which **8** predicts the labels. The victim model returns the labels to the client **9**, which can potentially use the returned labels to train its own model.

which initially were used in the security domain to prevent a server from Denial-of-Service attacks (DoS) [1], provide users with a small puzzle to be solved before giving access to the service (Dwork & Naor, 1993). In the case of Machine-Learning-as-a-Service (MLaaS), the answer to a query (e.g., the label predicted by a model) can similarly be released to a user once his or her computational or other system resources are spent. This can prevent a malicious user from obtaining enough examples labeled by a victim model to steal the model. Intuitively, our method shifts the trade-off between the quality of answers and robustness, that was introduced by previous defenses, to a trade-off between the computational cost and robustness to model stealing.

To minimize the impact of our defense on legitimate users, we tie the difficulty of the PoW problem (Back, 2020) to an estimate of how much information about a model has been extracted so far by a given user. We compute the privacy cost per query as an information leakage metric used in the differential privacy literature (Dwork et al., 2014). We choose to work with the privacy based metric because it is (1) agnostic to the learning task and model architecture, and (2) practical to compute for the model owner. We tune the per-query difficulty of the required proof-of-work based on the measured information gained by a user. PoW does not deteriorate the accuracy of answers and adds only a small increase (less than 2x) in the cost of querying public models for legitimate users. The cost is usually much higher (up to 100x) for attackers.

To summarize, our contributions are as follows:

- We introduce a new class of pro-active defenses against model stealing attacks that use proof-of-work mechanisms to adaptively increase the computational cost of queries with a negligible increase in the cost for legitimate users, and a high increase in cost for many attackers. **Our method is the first active defense strategy that leaves the model's accuracy entirely intact.**

- We propose a novel application of differential privacy budgets as a way to measure the amount of information leakage from answered queries. We calibrate the difficulty of the proof-of-work problem based on a user's query cost as estimated through its differential privacy budget.

- Our rigorous evaluation on four datasets and eight attacks validates that adversaries attempting to steal a model issue queries whose privacy cost accumulates up to 7x faster than for benign queries

---

[1]In a DoS attack, adversaries flood a server with superfluous requests, overwhelming the server and degrading service for legitimate users.

Table 1: **Taxonomy of model extraction attacks.** We use IND for in-distribution and OOD for out-of-distribution data. The MixMatch attack is the best performing one with very few queries run against the ML API and a high accuracy of the extracted model, mainly due to the abundant and efficient usage of the problem domain (i.e., task-specific) data. At the other end of the spectrum is DataFree attack with only synthetic OOD data.

| ATTACKS | DATA TYPE | # OF ATTACKER'S QUERIES | | | ADVERSARY'S GOAL | VICTIM RETURNS |
|---|---|---|---|---|---|---|
| | | MNIST | SVHN | CIFAR10 | | |
| MIXMATCH | PROBLEM DOMAIN | <10K | <10K | <10K | TASK ACCURACY | LABELS |
| ACTIVE LEARNING | PROBLEM DOMAIN | 1K - 10K | 1K - 25K | 1K - 9K | TASK ACCURACY | LABELS |
| KNOCKOFF | NATURAL OOD | 1K - 30K | 1K - 50K | 1K - 50K | TASK ACCURACY | LOGITS |
| COPYCAT CNN | NATURAL OOD | 1K - 50K | 1K - 50K | 1K - 50K | ACCURACY/FIDELITY | LABELS |
| JACOBIAN/-TR | LIMITED IND | 10K - 80 K | 10K - 150 K | 10K - 150 K | FIDELITY | LABELS |
| DATAFREE | SYNTHETIC OOD | 2M | 2M | 20M | TASK ACCURACY | LOGITS |

issued by legitimate users. Thus, our proof-of-work mechanism can deter attackers by increasing their computational cost significantly at a minimal overhead to legitimate users.

- We design adaptive attacks against our defense and show that they require simultaneous optimization of two opposing goals: (1) minimizing the privacy cost of queries run against the ML API and (2) maximizing the information leakage incurred by the queries. These conflicting targets render such attacks less effective, making them less attractive to adversaries.

## 2 BACKGROUND AND RELATED WORK

### 2.1 MODEL EXTRACTION ATTACKS

Model extraction aims to replicate some functionality of a victim model $f(x; \theta)$, where $x$ denotes an input and $\theta$ are parameters of the model. An attacker takes advantage of the ability to query the victim and observe the model's outputs, which are used to train a substitute model $f'(x; \theta')$. We test our defense against common forms of *fidelity* and *task accuracy* attacks. The *fidelity* extraction is given as input some target distribution $\mathcal{D}_T$ and a goal similarity function $S(p_1, p_2)$. It maximizes the target similarity between the victim and stolen models: $\max_{\theta'} \Pr_{x \sim \mathcal{D}_T} S(f'(x; \theta'), f(x; \theta))$. The Jacobian-based attack (Papernot et al., 2017b) is a high fidelity type of model extraction because it aims to align the gradients of the extracted model with the victim model's gradients. The *task accuracy* extraction aims at matching (or exceeding) the accuracy of the victim model. For the true task distribution $\mathcal{D}_A$, the goal is defined as: $\max_{\theta'} \Pr_{(x,y) \sim \mathcal{D}_A} [\arg\max(f'(x; \theta') = y]$. The task accuracy attacks include Knockoff Nets (Orekondy et al., 2019), DataFree (Truong et al., 2021), and MixMatch-based (Berthelot et al., 2019) model extraction.

We taxonomize these attacks in Table 1 from the perspective of a victim who observes the queries as well as the model's response on these queries. A defender wants to defend against any of these attacks, regardless of their intent, while not causing harm to legitimate users. Attackers with problem domain data need fewer queries and access to labels only is sufficient for them to steal a model (e.g., MixMatch extraction from Jagielski et al. (2020)). On the other hand, the lack of such data increases the number of required queries and might call for access to logits (Truong et al., 2021).

### 2.2 DEFENSES AGAINST MODEL EXTRACTION

We taxonomize the space of model defenses into three main classes. First, **active defenses** perturb the outputs from a defended model to harm training of the attacker's model. They trade-off defended model accuracy for better security. Second, **passive defenses** either truncate the outputs, which also lowers accuracy, or try to detect an attack, which makes them stateful since some form of information has to be stored about each user's queries and this might cause scalability issues. Other defenses are stateless. Third, **reactive defenses** try to prove a model theft rather than preventing the attack from happening. We propose a new class of defenses that are **pro-active**. We trade-off compute of the querying party for better security. We neither degrade the accuracy of the defended model nor require any changes to the serving model. Our defense is stateful but we only store privacy cost per user in a form of a single float number. Next, we describe specific examples of defenses from each class.

**Active defenses.** Lee et al. (2019) perturb a victim model's final activation layer, which preserves that top-1 class but output probabilities are distorted. Orekondy et al. (2020) modify the distribution of model predictions to impede the training process of the stolen copy by poisoning the adversary's gradient signal, however, it requires a costly gradient computation. The defenses proposed by Kariyappa & Qureshi (2020); Kariyappa et al. (2021) identify if a query is in-distribution or out-of-distribution (OOD) and then for OOD queries sends either incorrect or dissimilar predictions. The defenses introduced by Kariyappa et al. assume knowledge of attackers' OOD data that are generally hard to define a-priori and the selection of such data can easily bias learning (Hsu et al., 2020).

**Passive defenses.** PRADA by Juuti et al. (2019) and VarDetect by (Pal et al., 2021) are stateful and analyze the distribution of users' queries in order to detect potential attackers. The former assumes that benign queries are distributed in a consistent manner following a normal distribution while adversarial queries combine natural and synthetic samples to extract maximal information and have, e.g., small $\ell_2$ distances between them. The latter presumes that an outlier dataset, which represents adversarial queries, can be built incrementally and differs significantly from in-distribution data. However, PRADA does not generalize to attacks that utilize only natural data (Orekondy et al., 2019; Pal et al., 2020). Follow-up works (Atli et al., 2020) have specifically targeted model extraction attempts that use publicly available, non-problem domain natural data, but they assume access to data from such an attacker.

**Reactive defenses.** Watermarking allows a defender to embed some secret pattern in their model during training (Adi et al., 2018) or inference (Szyller et al., 2019). Jia et al. (2020a) proposed entangled watermarks to ensure that watermarks are not easy to remove. Dataset inference (Maini et al., 2021) identifies whether a given model was stolen by verifying if a suspected adversary's model has private knowledge from the original victim model's dataset. Proof of learning (Jia et al., 2021) involves the defender claiming ownership of a model by showing incremental updates of the model training. Unfortunately, reactive defenses are effective after a model theft and require a model owner to obtain partial or full access to the stolen model. This means that if an attacker aims to use the stolen model as a proxy to attack the victim model, or simply keeps the stolen model secret, it is impossible for the model owner to protect themselves with these defenses.

## 3 PROOF-OF-WORK AGAINST MODEL EXTRACTION

Our pro-active defense is based on the concept of proof-of-work (PoW) and requires users to expand some computation before receiving predictions. We reduce the impact on legitimate users by adjusting the difficulty of the required work based on the estimation of information leakage incurred by users' queries. The cost of queries is calculated using the privacy metric via the PATE framework (Papernot et al., 2017a; 2018), a standard approach to differential privacy in machine learning. Each user sends a query and receives a PoW puzzle to solve. The higher privacy cost incurred by a user, the more time it takes to solve the puzzle. The PoW server verifies whether the solution provided by the user is correct, and if the outcome is positive, the server releases original predictions.

### 3.1 THREAT MODEL

We assume that an attacker has black-box access to the victim model. That is, the attacker can only choose the query sent to the victim and observe the response in the form of a label or logits. The attacker has no access to the victim model parameters or training data. This is representative of most existing MLaaS systems. We use similar architectures for the attacker and victim as it is shown that attacks with shallower models can result in sub-optimal results (Juuti et al., 2019; Zhang et al., 2021). An attacker can have information about the strategy of our defense. For the purpose of our experiments, we assume that a legitimate user sends queries that are randomly selected from an in-distribution dataset unseen by the victim model and receives labels.

### 3.2 PROOF-OF-WORK PUZZLE

We propose to extend the scope of the PoW techniques to protect ML models from being easily stolen. One of the most popular proof-of-work (PoW) algorithms is HashCash (Back, 2002), which is used as a mining function in bitcoin (Back, 2020).

The PoW cost-function classification scheme presented in Back (2002) defines **six characteristics: 1)** efficiency of verifiability (0 - verifiable but impractical, 1/2 - practically verifiable, 1 - efficiently verifiable), **2)** type of the computation cost (0 - fixed cost (most desirable), 1/2 - bounded probabilistic cost, 1 - unbounded probabilistic cost), **3)** interactive or not, **4)** publicly auditable or not, **5)** does the server have a trapdoor in computing the function, **6)** parallelizable or not. For our purpose, the required properties are **1)** (with value $\geq 1/2$) and **3)**. The efficiency of verifiability has to be at least 1/2 to create a practical defense. The PoW-function for MLaaS has to be interactive **3)** since we want to control the difficulty of the PoW-function based on how much information leakage is incurred by a given user or globally after all answered queries. The remaining properties might be present or not. The HashCash function has unbounded probabilistic cost **2)**.

It is not necessary for the PoW cost-function to be publicly auditable **4)** if there is no conflict of interest and the service provider is not paid for more answered queries by an external 3-rd party, but directly by a user who sends a query. However, this property might be useful to verify how many queries a given server answered. It is not necessary but desirable that the server does not have a trapdoor **5)** (a shortcut in computing tokens) because it reduces complexity and administrable overhead (e.g., a requirement of a periodic key replacement). The non-parallelizable PoW cost-functions **6)** are of questionable practical value. In general, the interactive HashCash has a probabilistic cost (which in practice is acceptable) while not having the trapdoor, thus can be used as a PoW cost-function for MLaaS.

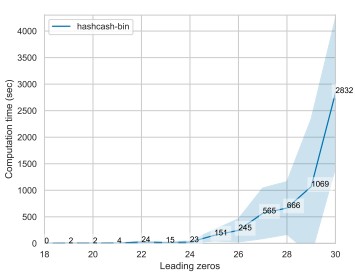

Figure 2: **Binary HashCash.**

With non-interactive PoW cost-functions, a client chooses its own challenge. In our case, we use the interactive version, where a server decides on the difficulty of the puzzle based on the information leakage incurred by a user. The puzzle in HashCash is to find a suffix that can be appended to a challenge string generated by the server (in the interactive mode) so that the message digest has a specific number of leading zeros. The levels of difficulty in the original implementation are rather coarse-grained since the hexadecimal representation is used. For the required minimal 5 leading zeros (in hexadecimal notation), the computation time is usually $< 1$ sec, for 6 leading zeros it can increase to 20 sec or more, and for 7 leading zeros it grows to even 511 seconds. Thus, we change the HashCash from comparing the leading zeros in the hexadecimal representation to the binary representation. One more zero in the hexadecimal format requires four zeros in the binary format. Hence, tracking zeros in the binary representation is more fine-grained and allows us to better control how much time is spent to solve a puzzle. This enables a tighter management of time spent by a given user on the computation before we reveal the final label.

In Figure 2, we present the dependency between the computation time and the required number of leading zero bits for the binary HashCash (aggregated across 8 runs). The average work time required is exponential in the number of zero bits and can be verified by executing a single hash (Nakamoto, 2008). The number of trials before a user achieves a *success* is modeled by a geometric distribution, which is the probability distribution over the number of required Bernoulli trials until the *success*. The expected number of trials for $k$ leading zeros is $2^k$, its variance is $2^k(2^k - 1)$, and an approximate standard deviation is $2^k$ (usually $2^k \gg 1$). The official implementation is non-interactive, coarse-grained, and written in Python 2.3 (Back, 2020). We extend the original version of the code and implement the HashCash cost-function, in both interactive and non-interactive modes, make it fine-grained, and port it to Python 3.8.

### 3.3 Calibration of Puzzle Difficulty

After specifying the proof-of-work method, we use it to generate puzzles for a given user. We calibrate the difficulty of PoW puzzles which is characterized by the number of leading zeros in the message digest that needs to be returned by the client to demonstrate they solved the puzzle. Our calibration is based on the cost of each query: the cost should reflect the information leakage about the training data as a result of the answers released for queries. Thus, we naturally turn to literature on reasoning about privacy in ML, and in particular differential privacy.

**Evaluating the cost of queries with differential privacy.** We define model information leakage as the amount of useful information a user is able to gain from a set of queries run against a victim model.

We use the information leakage based on the user's queries to inform our defense and ultimately help limit the amount of information that malicious users can gain. Since our defense works on a per user basis, it requires the identification of users. Our main method is to measure the information leakage based on the privacy budget incurred by a given user, which is a known technique from the differential privacy literature (Dwork et al., 2014). There are two canonical approaches to obtaining privacy in deep learning: DP-SGD and PATE. DP-SGD is not applicable in our case as it is an algorithm level mechanism used during training, which does not measure privacy leakage for individual test queries. On the other hand, PATE is an output level mechanism, which allows us to measure per-query privacy leakage. PATE creates teacher models using the defender's training data and predictions from each teacher are aggregated into a histogram where $n_i$ indicates the number of teachers who voted for class $i$. The privacy cost is computed based on the consensus among teachers and it is smaller when teachers agree. The consensus is high if we have a low probability that the label with maximum number of votes is not returned. This probability is expressed as: $\frac{1}{2} \sum_{i \neq i^*} \text{erfc}(\frac{n_{i^*} - n_i}{2\sigma})$, where $i^*$ is an index of the correct class, erfc is the complementary error function, and $\sigma$ is the privacy noise (see details in Appendix D). Then a student model is trained by transferring knowledge acquired by the teachers in a privacy-preserving way. This student model corresponds to the attacker's model trained during extraction. Papernot et al. (2017a) present the analysis (e.g., Table 2) of how the performance (in terms of accuracy) of the student model increases when a higher privacy loss is incurred through the answered queries.

**From the privacy cost to the puzzle difficulty.** The reference cost is for a standard user who is assumed to label in-distribution data that are sent to the server in a random order. The goal is to find a mapping from a user's privacy cost to the number of leading zeros using the available data: (1) privacy cost, number of queries, and timing for legitimate users, and (2) timing to solve a puzzle for a number of leading zeros. Based on the data from legitimate users, we create a model that predicts the total privacy cost from the number of queries to the privacy cost. This first model extrapolates to any number of queries. The more data on legitimate users we have access to, the more precise the specification of the dependency between the number of queries and the cumulative privacy cost. We interpolate the second dataset by creating a linear model that predicts the number of leading zeros for the desired time to solve the puzzle. We take the logarithm of the PoW time, which transforms this exponential dependency to a linear one. To allow for the additional overhead of PoW, we set a legitimate user's total timing to be up to 2x longer than the initial one. The query cost and timing are limited for a legitimate user in comparison to an attacker and so the models allow us to handle arbitrarily small or large privacy costs and timings.

**Answer queries with PoW.** When a user sends a new batch of queries for prediction, the first model maps from the user's query number to the expected accumulated query cost for a legitimate user. We take the absolute difference $x$ between the actual total query cost of the user and the predicted legitimate query cost. If the costs for legitimate users differ from each other substantially then, instead of using difference to a single baseline, we could define upper and lower bounds for the expected privacy costs. Next, we map from the difference $x$ to the timing for the PoW using, for example, an exponential function: $f(x) = a^x$, where $a$ is a parameter set by a model's owner. In our experiments $a = 1.0075$ so that the standard user's cost does not increase beyond 2x and the cost for the attackers is maximized. A legitimate user should stay within a low regime of the difference and an attacker's difference should grow with more queries thus requesting more work within PoW. Then, we use the second model to map from the desired timing of PoW to the number of leading zero bits that have to be set for the puzzle. We find that using a simple linear regression is sufficient for both models.

## 4 EMPIRICAL EVALUATION

We evaluate our defense against different types of model extraction attacks. We show that our privacy-based cost metric is able to differentiate between the information leakage incurred by attackers vs legitimate users (Figure 3). The accuracy gain as measured on an adversary's model against a victim's privacy loss assesses how efficiently our defense can counter a given attack. Our results show that no adversary can achieve a substantial gain over a user with legitimate queries (Figure 4). Finally, we perform an end-to-end evaluation (Table 2), which indicates that run time for a legitimate user is only up to 2x higher whereas it is substantially increased (more than 100x) for most attackers.

## 4.1 EXPERIMENTAL SETUP

We use MNIST, FashionMNIST (in the Appendix), SVHN, and CIFAR10 datasets to test our defense. To evaluate the accuracy of the extracted model, a subset of the corresponding test set is used, which does not overlap with an attacker's queries. In our experiments, for MNIST and FashionMNIST, we use a victim model with the MnistNet architecture (shown in B.2) and test set accuracies of 99.4% and 92.2% respectively. ResNet-34 with test set accuracies of 96.2% and 95.6% is used for SVHN and CIFAR10, respectively. Next, we describe the model stealing attacks used in our experiments.

**Jacobian** and Jacobian-TR (targeted) attacks use 150 samples from the respective test sets for the initial substitute training set. A value of $\lambda = 0.1$ is used for the dataset augmentation step and the attacker model is trained for 20 epochs with a learning rate of 0.01 and momentum of 0.9 after each augmentation step.

**DataFree** is run in the same way and with the same parameters as in the original implementation from Truong et al. (2021). Specifically for MNIST and SVHN, we use a query budget of 2M while for CIFAR10, we use a budget of 20M. The attacker's model is trained using ResNet-18 in all cases.

**Entropy** (Active Learning) based attack uses an iterative training process where 1000 queries are made based on the entropy selection method. This is followed by the attacker model being trained for 50 epochs with a learning rate of 0.01 and momentum of 0.9 on all queries that have been asked till that point. We then repeat this process.

**MixMatch** based attack queries a victim model using examples randomly sampled from a test set. Active learning approaches were tried but had negligible effect on the accuracy of the stolen model. After the labeling, the same number of additional unlabeled samples are taken from the test set. The attacker's model is trained using Wide-ResNet-28 (Oliver et al., 2018) for all datasets and using 128 epochs. The other hyperparameters are kept the same as in the original paper (Berthelot et al., 2019).

**Knockoff** queries a model from an out-of-distribution dataset, in our case it is Imagenet (Deng et al., 2009) dataset for MNIST, while SVHN and CIFAR100 are used for CIFAR10. We use the *Random* and *Adaptive* querying strategies in our experiments. After the querying process has been completed, an attacker model is trained for 100 epochs with a learning rate of 0.01 and momentum of 0.9.

**CopyCat** considers two cases, namely where the attacker has access to problem domain data and where the attacker has access to non-problem domain data (Correia-Silva et al., 2018). We only consider the case where the attacker has access to non-problem domain data (in this case ImageNet) as the other case is the same method as our standard (random) querying.

**EntropyRev** and **WorstCase** are adaptive attacks designed to stress test our defense.

Since the attacks considered originally had highly different setups, we made adjustments to unify them. For example, for MixMatch, we use test sets to select queries as opposed to train sets because other attacks assume no access to the train sets. For the active learning attacks, or Jacobian approaches, the number of epochs and hyperparameters used for training are kept the same. We train the models starting from a random initialization.

## 4.2 INFORMATION LEAKAGE

We compare the privacy cost measured on the victim model for different attacks. To compute the privacy cost, we use PATE (Papernot et al., 2018) with 250 teachers for MNIST and SVHN, and 50 teachers for CIFAR10. The scale of the Gaussian noise $\sigma$ is set to 10 for MNIST and SVHN, and to 2 for CIFAR10. Results for *Standard* query selection represent a legitimate user. From Figure 3, we find that the privacy metric correctly shows higher values for the attackers compared to a *Standard* user. Privacy cost increases much more for DataFree, CopyCat, and Knockoff attacks that use OOD data, compared to the attacks that use in-distribution data. The Jacobian attack starts from a small seed of in-distribution samples and then perturbs them using the Jacobian of the loss function and therefore shows lower cost values. However, despite its lower privacy cost, the Jacobian attack achieves much lower accuracy of the extracted model than other attacks (Figure 4).

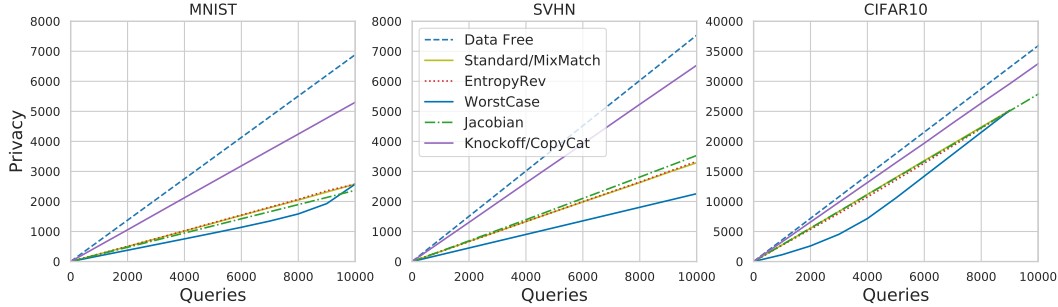

Figure 3: **Privacy Cost vs Number of Queries.** We measure the privacy cost against the number of queries for various query selection strategies on the MNIST, SVHN and CIFAR10 datasets.

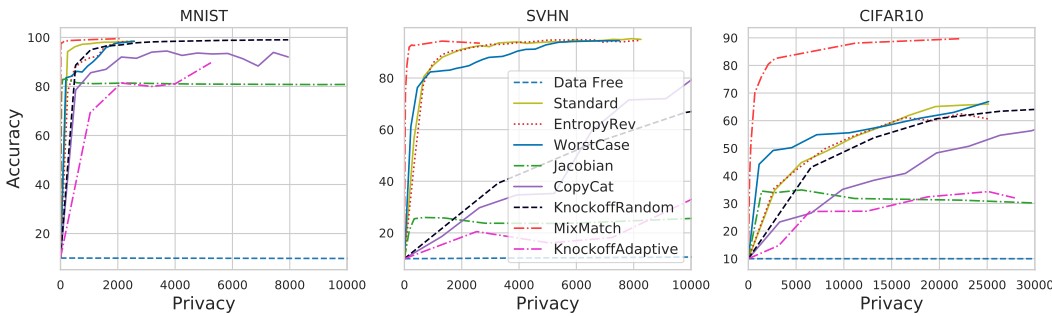

Figure 4: **Accuracy vs Privacy Cost.** We measure the accuracy against the privacy cost for various query selection strategies on the MNIST, SVHN and CIFAR10 datasets. An attacker cannot achieve a better tradeoff than a user with in-distribution data.

### 4.3 PROOF-OF-WORK AGAINST ATTACKS

The execution time for a legitimate user is longer by less than 2x after our defense is applied while the execution times of Knockoff and DataFree attacks are up to 100x longer (see Table 2 and Figure 8). The 2x longer execution is chosen as an acceptable difference for legitimate users that makes the attacks significantly more expensive. The cost of execution of the DataFree attack is totally dominated by PoW due to its high privacy cost incurred by many queries with almost random content. This is because this attack has no access to in distribution data.

In fact, the results for MixMatch in Figure 4 and Table 2 show how valuable is the access to in-distribution data combined with semi-supervised learning. The cost of querying the victim model is much lower while the task accuracy of the stolen model is substantially higher than for other attacks. Nevertheless, the MixMatch extraction requires unlabeled in-distribution samples, which might not be feasible to collect for the adversary.

For attacks consisting of synthetic queries, such as DataFree, even if an attacker attempts to reduce their cost, the usage of the synthetic data, which are out-of-distribution, causes our metrics to increase much faster than for legitimate users and thus this approach is not able to bypass our defense. Furthermore, trying to minimize the cost for an attack involving synthetic data requires the attacker to perform more computation because there is no set of examples from which they can choose to query and so instead a larger set of possible queries needs to be generated.

### 4.4 ADAPTIVE ATTACKS

We design and evaluate attackers who are aware of our defense and actively try to adapt to it. Their goal is to minimize the privacy cost calculated by the victim while maximizing the accuracy of the stolen model. **EntropyRev** attack uses *entropy* to approximate the cost computed by a victim and selects queries with minimal instead of maximal entropy, which is a *reversed* strategy to the entropy-based active learning attack. The achieved entropy cost (and also the privacy and gap) is

Table 2: **Comparison of the attacks against the PoW-based defense.** The notation: IND- in-distribution, SYN- Synthetic. IND [†] denotes limited in-distribution data with additional data generated using the Jacobian augmentation. MixMatch[‡] uses as many additional unlabeled training examples for the semi-supervised training as the number of answered queries. The results for Standard, EntropyRev, and WorstCase are the same so we report only Standard. Time PoW is after applying our defense.

| DATASET | ATTACK | DATA TYPE | # OF QUERIES | PRIVACY PATE | TIME (SEC) | TIME PoW | ACCURACY (%) | FIDELITY (%) |
|---|---|---|---|---|---|---|---|---|
| MNIST | MIXMATCH | IND | 8K[‡] | 2047 | 58.9 | 67.1 | 99.5 | 99.6 |
| MNIST | STANDARD | IND | 10K | 2576 | 73.6 | 83.9 | 98.4 | 98.7 |
| MNIST | KNOCKOFFRANDOM | IMAGENET | 10K | 5301 | 73.6 | 21283 | 98.7 | 99.0 |
| MNIST | KNOCKOFFADAPTIVE | IMAGENET | 10K | 5251 | 73.6 | 21235 | 89.7 | 90.0 |
| MNIST | COPYCAT | IMAGENET | 10K | 5301 | 73.6 | 21283 | 92.7 | 92.9 |
| MNIST | JACOBIAN | IND [†] | 10K | 2421.6 | 73.6 | 91.1 | 81.4 | 81.5 |
| MNIST | DATAFREE | SYN | 2M | 1.39E6 | 14375 | 276904 | 95.18 | 95.7 |
| SVHN | MIXMATCH | IND | 8K[‡] | 2629 | 277.8 | 286.3 | 97.2 | 97.4 |
| SVHN | STANDARD | IND | 24K | 7918 | 833.3 | 859.4 | 95.6 | 95.3 |
| SVHN | KNOCKOFFRANDOM | IMAGENET | 24K | 15656 | 833.3 | 33413 | 92.5 | 92.5 |
| SVHN | KNOCKOFFADAPTIVE | IMAGENET | 24K | 15195 | 833.3 | 33372 | 43.9 | 44.7 |
| SVHN | COPYCAT | IMAGENET | 24K | 15656 | 833.3 | 33413 | 87.6 | 87.8 |
| SVHN | JACOBIAN | IND [†] | 24 K | 8763 | 833.3 | 6424 | 26.0 | 26.6 |
| SVHN | DATAFREE | SYN | 2M | 1.52E6 | 68355 | 330769 | 95.3 | 97.2 |
| CIFAR10 | MIXMATCH | IND | 8K[‡] | 22378 | 131.8 | 163 | 91.7 | 92.2 |
| CIFAR10 | STANDARD | IND | 9K | 25147 | 148.3 | 182.4 | 65.7 | 66.0 |
| CIFAR10 | KNOCKOFFRANDOM | CIFAR100 | 9K | 29584 | 148.3 | 23921 | 64.0 | 64.2 |
| CIFAR10 | KNOCKOFFADAPTIVE | CIFAR100 | 9K | 27927 | 148.3 | 23528 | 32.0 | 32.3 |
| CIFAR10 | COPYCAT | CIFAR100 | 9K | 29584 | 148.3 | 23921 | 54.2 | 54.8 |
| CIFAR10 | JACOBIAN | IND [†] | 9K | 25574 | 148.3 | 199.9 | 30.5 | 31.1 |
| CIFAR10 | KNOCKOFF | CIFAR100 | 30K | 98355 | 494.4 | 86185 | 87.8 | 88.7 |
| CIFAR10 | COPYCAT | CIFAR100 | 30K | 98355 | 494.4 | 86185 | 78.1 | 77.7 |
| CIFAR10 | DATAFREE | SYN | 20M | 7.14E7 | 3.22E5 | 2.95E6 | 87.15 | 88.0 |

lower than for a standard (random) querying, however, the accuracy of the stolen model also drops in most cases (Figure 5 in Appendix). **WorstCase** attack uses in-distribution data (we also test it on OOD data in Figure 6) and has access to (or can infer) the exact victim's privacy cost to minimize the cost of adversarial queries. Privacy cost does not differ much between in-distribution queries so this attacker cannot game our system.

A more realistic scenario of a powerful adaptive attacker is that a victim also releases scores (e.g., softmax outputs) that are used to obtain an estimation of the victim's privacy cost. However, the estimation of the cost is known only after the query is answered, thus it cannot be used by the attacker to order the queries sent for prediction. Another adaptive adversary can attack our operation model with multiple fake accounts instead of funneling all their queries through a single account or there can be multiple colluding users. Our approach not only defends against this because the attacker still needs to solve the proof-of-work for each of the user accounts, but it also simplifies the task of identifying colluding users. We note that one appealing aspect of our defense is that it relies on the privacy cost that can easily be summed across users because of the composition property of differential privacy. We leave the implementation and assessment of this idea for future work.

## 5 CONCLUSIONS

Model extraction is a threat to the confidentiality of machine learning models and can be used as reconnaissance prior to mounting other attacks. It can leak private datasets that are expensive to produce and contain sensitive information. We propose the first pro-active defense against model stealing attacks that leaves the model's accuracy entirely intact. We wrap original model predictions in a proof-of-work (PoW) to force users to expend additional compute on solving a PoW puzzle before they can read the output predicted by the model. To minimize the impact on legitimate users, we tie the difficulty of the puzzle to an estimate of information leakage per user's account. We show that the information leakage incurred by users' queries as measured by the privacy cost using PATE is a reliable indicator of which user tries to extract more information from a model than expected. We thus believe that our approach can be successfully applied by practitioners to increase the security of models which are already publicly deployed.

## 6 ETHICS STATEMENT

One possible ethical concern regarding our defense is that the application of Proof-of-Work (PoW) requires additional computation, which could increase energy usage and thus also the $CO_2$ emission. However, since the increase in computation is very little for a benign user, the effect of PoW will be minimal whereas for an attacker, extra compute to train a model, often also involving the use of GPUs, is required and this outweighs any effect of the computation required for our PoW based defense.

To reduce the energy wastage, our approach can be adapted to rely on PoET (Proof-of-Elapsed-Time), for example, as implemented by Hyperledger Sawtooth, instead of PoW. The users' resource (e.g., a CPU) would have to be occupied for a specific amount of time without executing any work. At the end of the waiting time, the user sends proof of the elapsed time instead of proof of work. Both proofs can be easily verified. PoET reduces the potential energy wastage in comparison with PoW but requires access to a specialized hardware, namely new secure CPU instructions that are becoming widely available in consumer and enterprise processors. If a user does not have such hardware on premise, the proof of elapsed time could be produced using a service exposed by a cloud provider (e.g., Azure VMs that feature TEE [2]).

In our defense, a legitimate user's cost (2X) is not prohibitive but also non-negligible. Previous work (Orekondy et al., 2020; Kariyappa & Qureshi, 2020) and our results suggest that there is *no free lunch* - the active defenses against model extraction either sacrifice the accuracy of the returned answers or force users to do more work. The principal strength of our active defense approach is that it does not sacrifice accuracy, which comes at the expense of computational overhead.

Another aspect is that users with more powerful hardware (e.g., CPUs with a higher frequency) could compute the puzzle from PoW faster than other users with less performant chips. We note that this is directly related to the design of PoW cost-functions. For example, HashCash is difficult to parallelize, however, Back (2002) argues that protection provided by non-parallizable PoW cost-functions is marginal since an adversary can farm out multiple challenges as easily as farm out a sub-divided single challenge.

## 7 REPRODUCIBILITY STATEMENT

We submit our code in the supplementary material. In the *README.md* file we provide the main commands needed to run our code. We also describe the pointers to the crucial parts of the code that directly reflect the implementation described in the main part of the submission. We describe the attacks used in our experiments thoroughly in the Appendix in Section A.1.

## ACKNOWLEDGMENTS

We would like to acknowledge our sponsors, who support our research with financial and in-kind contributions: DARPA through the GARD project, Microsoft, Intel, CIFAR through the Canada CIFAR AI Chair and AI catalyst programs, NFRF through an Exploration grant, and NSERC COHESA Strategic Alliance. Resources used in preparing this research were provided, in part, by the Province of Ontario, the Government of Canada through CIFAR, and companies sponsoring the Vector Institute https://vectorinstitute.ai/partners. Finally, we would like to thank members of CleverHans Lab for their feedback.

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

Table 3: **Comparison of the attacks against the PoW-based defense.** (Extended results from Table 2.) We present the number of required queries, gap, entropy, privacy cost PkNN (as computed for alternative metrics from Appendix B.6), time before (Time) and after PoW (Time PoW), and accuracy (ACC) for different datasets and attacks. The results are aligned according to the highest accuracy achieved by the attacks.

| DATASET | ATTACK | DATA TYPE | # OF QUERIES | GAP | ENTROPY | PRIVACY PkNN | TIME (SEC) | TIME PoW | ACC (%) |
|---|---|---|---|---|---|---|---|---|---|
| MNIST | MixMatch | IND | $100^{\ddagger}$ | 0.17 | 0.19 | 0.36 | 0.69 | 0.97 | 98.0 |
| MNIST | MixMatch | IND | $1K^{\ddagger}$ | 4.51 | 2.8 | 1.01 | 7.2 | 8.61 | 98.9 |
| MNIST | MixMatch | IND | $4K^{\ddagger}$ | 21.4 | 13.1 | 2.55 | 28.8 | 34.8 | 99.2 |
| MNIST | MixMatch | IND | $8K^{\ddagger}$ | 46.3 | 28.7 | 4.04 | 58.8 | 71.3 | 99.5 |
| MNIST | Standard | IND | 10K | 59.7 | 37.5 | 4.58 | 73.4 | 88.92 | 98.4 |
| MNIST | Knockoff | ImageNet | 10K | 4539 | 4367 | 50 | 73.4 | 42513 | 98.7 |
| MNIST | CopyCat | ImageNet | 10K | 4539 | 4367 | 50 | 73.4 | 42513 | 93.2 |
| MNIST | Jacobian | IND $^{\dagger}$ | 10K | 41.9 | 46.2 | 3.46 | 73.4 | 88.9 | 82.4 |
| MNIST | DataFree | SYN | 2M | 1747972 | 1915677 | 5367 | 14852 | 1.1E9 | 95.18 |
| MNIST | EntropyRev | ImageNet | 10K | 3750 | 3295 | 62.2 | 73.4 | 58078 | 91.03 |
| MNIST | WorstCase | ImageNet | 10K | 444.6 | 704 | 34 | 73.4 | 24474 | 95.3 |
| SVHN | MixMatch | IND | $250^{\ddagger}$ | 8.95 | 10.45 | 0.96 | 2.19 | 2.64 | 90.8/95.82* |
| SVHN | MixMatch | IND | $1K^{\ddagger}$ | 30.3 | 35.9 | 1.84 | 8.8 | 10.6 | 96.8/96.87* |
| SVHN | MixMatch | IND | $4K^{\ddagger}$ | 138.4 | 156.5 | 4.20 | 35 | 41.2 | 97.3/97.07* |
| SVHN | MixMatch | IND | $8K^{\ddagger}$ | 280.1 | 314.9 | 6.39 | 70 | 82.5 | 97.2 |
| SVHN | Standard | IND | 25K | 868 | 975 | 15.3 | 218.9 | 257.4 | 94.7 |
| SVHN | Knockoff | ImageNet | 25K | 10594 | 10918 | 168 | 218.9 | 247506 | 92.5 |
| SVHN | CopyCat | ImageNet | 25K | 10594 | 10918 | 168 | 218.9 | 247506 | 87.6 |
| SVHN | Jacobian | IND $^{\dagger}$ | 25K | 1411 | 1429 | 23.0 | 218.9 | 5121 | 44.6 |
| SVHN | DataFree | SYN | 20M | 795382 | 790731 | 12919 | 17443 | 1.80E9 | 95.3 |
| SVHN | EntropyRev | ImageNet | 25K | 10467 | 10771 | 171.4 | 218.9 | 224868 | 86.3 |
| SVHN | WorstCase | ImageNet | 25K | 5789 | 6907 | 26.1 | 218.9 | 254.1 | 88.3 |
| CIFAR10 | MixMatch | IND | $250^{\ddagger}$ | 7.41 | 5.80 | 0.8 | 1.80 | 2.24 | 70.4/87.98* |
| CIFAR10 | MixMatch | IND | $1K^{\ddagger}$ | 28.24 | 21.78 | 1.55 | 7.2 | 8.96 | 83.8/90.63* |
| CIFAR10 | MixMatch | IND | $4K^{\ddagger}$ | 118.23 | 89.33 | 3.44 | 28.8 | 35.25 | 89.1/93.29* |
| CIFAR10 | MixMatch | IND | $8K^{\ddagger}$ | 231.88 | 178.40 | 5.25 | 58.8 | 70 | 91.7 |
| CIFAR10 | Standard | IND +TinyImages | 30K | 704.9 | 588.3 | 15.9 | 215.6 | 262.5 | 89.6 |
| CIFAR10 | Knockoff | CIFAR100 | 30K | 6889 | 5656 | 146 | 215.6 | 290377 | 87.8 |
| CIFAR10 | CopyCat | CIFAR100 | 30K | 6889 | 5656 | 146 | 215.6 | 290377 | 75 |
| CIFAR10 | Jacobian | IND $^{\dagger}$ | 10K | 1060 | 783 | 23.1 | 71.9 | 2165 | 28.2 |
| CIFAR10 | DataFree | SYN | 20M | 7.65E8 | 6.84E6 | 146540 | 143612 | 2.29E11 | 87.15 |
| CIFAR10 | EntropyRev | CIFAR100 | 30K | 7003 | 5675 | 148.9 | 215.6 | 290672 | 70.9 |
| CIFAR10 | WorstCase | CIFAR100 | 30K | 4407 | 3442 | 91.8 | 215.6 | 139208 | 70.5 |

Table 4: **Taxonomy of model extraction defenses.**

| DEFENSES | DEFENSE TYPE | METHOD | STATE | ACCURACY |
|---|---|---|---|---|
| PRADA | PASSIVE | COMPARE DISTRIBUTIONS | STATEFUL | PRESERVED |
| VarDetect | PASSIVE | COMPARE DISTRIBUTIONS | STATEFUL | PRESERVED |
| Prediction Poisoning | ACTIVE | PERTURB OUTPUTS | STATELESS | BOUNDED |
| Adaptive Misinformation | ACTIVE | PERTURB OUTPUTS | STATELESS | BOUNDED |
| Watermarking | REACTIVE | EMBED SECRET | N/A | N/A |
| Dataset Inference | REACTIVE | RESOLVE OWNERSHIP | N/A | N/A |
| Proof of Learning | REACTIVE | TRAINING LOGS | N/A | N/A |
| PoW | PRO-ACTIVE | PoW + PATE | STATEFUL | PRESERVED |

# A  MODEL EXTRACTION ATTACKS

## A.1  DETAILED DESCRIPTION OF ATTACKS USED

The latest attacks that we implement and check against using our methods are as follows:

1. **Jacobian** (Jacobian matrix based Dataset Augmentation), which was originally proposed by Papernot et al. (2017b), is an attack that uses synthetic data to generate queries. The substitute training process involves querying for the labels in the substitute training set $S_p$, training a model and then using augmentation on the substitute training set $S_p$ to generate $S_{p+1}$. For each sample $x_i \in S_p$, we generate a new synthetic sample $x'_i \in S_{p+1}$ by using the sign of the substitute

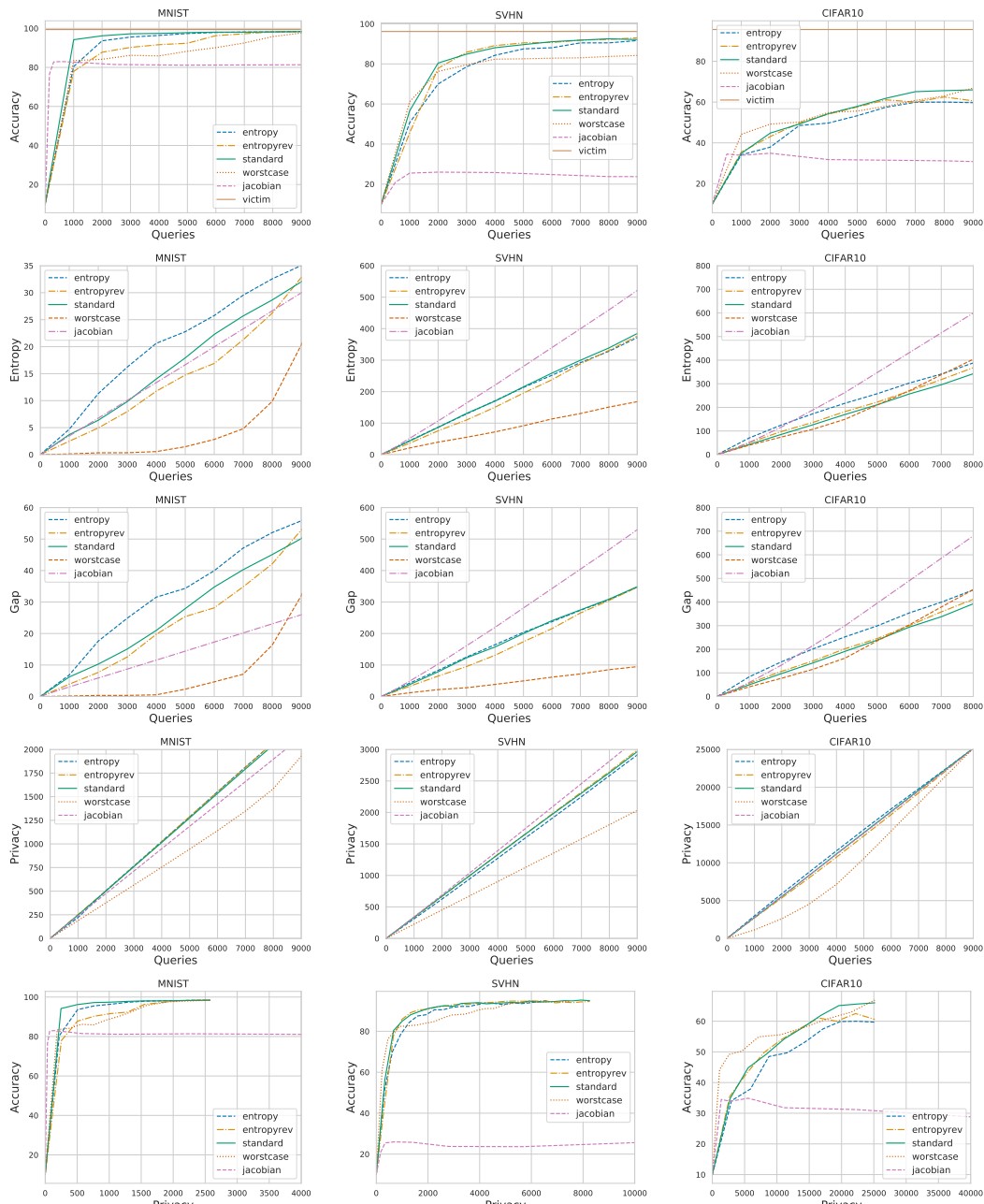

Figure 5: **Adaptive attacks using in-distribution data.**. We observe a tradeoff between attackers' goal to bypass our defense and the resulting accuracy they achieve. Accuracy, Entropy, Gap, and Privacy costs of the victim model vs number of Queries and Accuracy vs Privacy for MNIST, SVHN and CIFAR10 datasets. We note that for the Accuracy vs Privacy graph, the final privacy cost of all methods is the same. This is because at the end of the querying process all methods have used all the available samples.

DNN's Jacobian matrix dimension $J_{F(x)}$ corresponding to the label assigned to input $x_i$ by the oracle $O$.: $x'_i = x_i + \lambda \cdot \text{sgn}(J_F[O(x_i)])$. Note that the set $S_{p+1}$ is the union of $S_p$ and the new points generated. These steps are then repeated. The aim of the attack is to identify directions in which the model's output is varying around an initial set of points so that the decision boundary can be approximated. Jacobian is used as a model extraction attack, however, it was designed

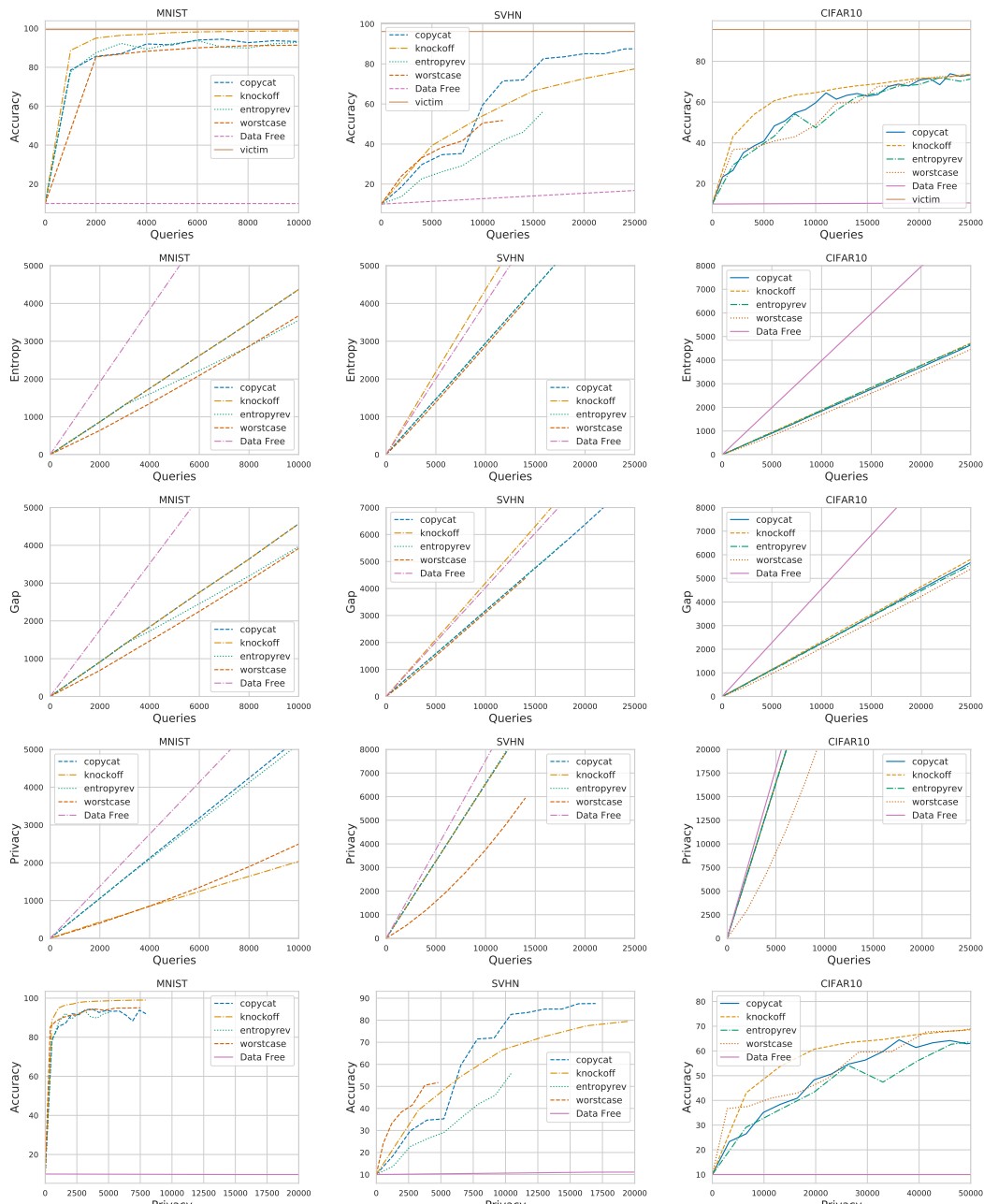

Figure 6: **Adaptive attacks using out-of-distribution data.** Accuracy, Entropy, Gap, and Privacy of the victim model vs number of Queries and Accuracy vs Privacy for the MNIST, SVHN and CIFAR10 datasets.

to optimize for the transferability of adversarial examples between models, which is a different objective. This is called a reconnaissance-motivated adversary who is more interested in a high fidelity extraction reconstruction whereas a theft-motivated adversary optimizes for a high accuracy extraction (Jagielski & Papernot, 2020).

2. **Jacobian-TR** is a variation of Jacobian proposed by Juuti et al. (2019). Here a new sample $x'_i \in S_{p+1}$ is generated from an existing sample $x_i \in S_p$ using the sign of the Jacobian matrix dimension for a random target class as: $x'_i = x_i - \lambda \cdot \text{sgn}(J_F[y_T])$, where $y_T$ is a random class not equal to the oracle's prediction $O(x_i)$.

3. **MixMatch** is a semi-supervised learning approach, which was originally proposed by Berthelot et al. (2019) and applied by Jagielski et al. (2020) for high fidelity and accuracy extraction. This model extraction method shows that is is possible to steal a model even with a small number of labeled examples (Jagielski et al., 2020). This attack uses an initial set of labeled data points along with an equally sized set of unlabeled examples. The algorithm produces a processed sample-label pairs for both the labeled and unlabeled data points through steps involving data augmentation (random horizontal flips and crops) that leave class semantics unaffected, label guessing (average prediction over augmentations), sharpening (to produce lower entropy predictions), and MixUp (a linear interpolation of two examples and their labels). The ablation study indicates that MixUp (especially on unlabeled points) and sharpening are the most crucial steps. This model is trained using a standard semi-supervised loss for the processed batches, which is a linear combination of the cross-entropy loss used for labeled examples and Brier's score (squared $\ell_2$ loss on predictions) (Brier et al., 1950) for the unlabeled examples.

4. **DataFree** (Data Free Model Extraction) is an attack proposed by Truong et al. (2021) that does not require a surrogate dataset and instead uses a strategy inspired by Generative Adversarial Networks (GANs) to generate queries. A generator model creates input images that are difficult to classify while the student model serves as a discriminator whose goal during training is to match the predictions of the victim (oracle) on these images. The student and generator act as adversaries, which respectively try to minimize and maximize the disagreement between the student and the victim. DataFree assumes access to the softmax values from the victim model and requires gradient approximation to train the generator, thus many queries are needed to be issued, e.g., in the order of millions when attacking models trained on SVHN or CIFAR10 datasets.

5. **AL** (Active Learning) methods for model extraction help the attacker to select a set of points from a pool of in-distribution unlabeled examples. The active learning methods serve as heuristics that are used to compute scores for queries based on the trained substitute model (a.k.a. stolen model). The attacker uses obtained labels to improve the substitute model and can iteratively repeat the selection and query process. AL methods assume that an unlabeled in-distribution dataset $d$ and a classification model with conditional label distribution $P_\theta(y \,|\, x)$ are provided. We implement the following heuristics from the active learning literature, particularly from Pal et al. (2019):

   - *Entropy*: select points which maximize the entropy measure:
     $x^* = \arg\max_{x \in d} -\sum_i P_\theta(y_i \,|\, x) \log P_\theta(y_i \,|\, x)$, where $y_i$ ranges over all possible labels.
   - *Gap*: select points with smallest gap in probability of the two most probable classes:
     $x^* = \arg\min_{x \in d} P_\theta(\hat{y}_1 \,|\, x) - P_\theta(\hat{y}_2 \,|\, x)$, where $\hat{y}_1$ and $\hat{y}_2$ are respectively the most and second most probable classes for $x$, according to the currently extracted model.
   - *Greedy*: run $k$-nearest neighbour algorithm by selecting $k$ labeled points as cluster centers that minimize the furthest distance from these cluster centers to any data point. Formally, this goal is defined as $\min_{\mathcal{S}:|\mathcal{S} \cup \mathcal{D}| \leq k} \max_i \min_{j \in \mathcal{S} \cup \mathcal{D}} \Delta(\mathbf{x}_i, \mathbf{x}_j)$, where $\mathcal{D}$ is the current training set and $\mathcal{S}$ is our new chosen center points. This definition can can be solved greedily as shown in (Sener & Savarese, 2017).
   - *DeepFool*: select data points, using the DeepFool attack proposed by (Moosavi-Dezfooli et al., 2016), that require the smallest adversarial perturbation to fool the currently extracted model. Intuitively, these examples are the closest ones to decision boundaries. The victim model is queried with such unperturbed examples.

   The attacks that use active learning methods include ActiveThief (Pal et al., 2020) and model extraction with active learning (Chandrasekaran et al., 2020).

6. **Knockoff Nets** is a two step approach to model functionality stealing proposed by Orekondy et al. (2019). The first step involves querying a set of input images to obtain predictions (softmax output), followed by the second step where a *knockoff* (illegal copy of the original model) is trained with the queried image-prediction pairs. Two strategies are presented for the querying process, namely a *Random* strategy where the attacker randomly samples queries (without replacement) from a different distribution compared to the victim's training set and a novel *Adaptive* strategy where a policy is learnt based on reinforcement learning methods (e.g., gradient bandit algorithm) to improve the sample efficiency of queries and aid the interpretability of the black-box victim model.

7. **CopyCat CNN** is a method proposed by Correia-Silva et al. (2018) which aims to extract a model using random non labeled data. This attack also consists of two steps where a fake dataset is first

generated using random queries to the model which is then followed by CopyCat training. The paper considers two cases, namely where the attacker has access to problem domain data and where the attacker has access to non problem domain data. We only consider the case where the attacker has access to non problem domain data (in this case ImageNet) as the other case is the same method as our standard (random) querying.

Another attack related to the above methods is EAT which we briefly describe below:

1. **Extended Adaptive Training (EAT)** is an extraction strategy applied by Chandrasekaran et al. (2020) that relies on active learning methods. The training of an attacker's model is similar to the approach we use for our active learning methods with an iterative process that retrains a model on the newest generated points. The exact strategy used for the SVM-based models is active selection. This approach selects to query the point that is the closest to a decision boundary.

## B  ADDITIONAL EXPERIMENTAL RESULTS

### B.1  DETAILS ON THE EXPERIMENTAL SETUP

Our experiments were performed on machines with Intel®Xeon®Silver 4210 processor, 128 GB of RAM, and four NVIDIA GeForce RTX 2080 graphics cards, running Ubuntu 18.04.

### B.2  MNISTNET ARCHITECTURE

The architecture used for MNIST dataset consists of two convolutional layers followed by two fully connected layers. Below is the detailed list of layers in the architecture used (generated using *torchsummary*).

```
LeNet style architecture for MNIST:
----------------------------------------------------------------
        Layer type              Output Shape          Param #
================================================================
        Conv2d-1            [-1, 20, 24, 24]              520
        ReLU-2
        MaxPool2d-3
        Conv2d-4             [-1, 50, 8, 8]            25,050
        ReLU-5
        MaxPool2d-6
        Linear-7                  [-1, 500]           400,500
        ReLU-8
        Linear-9                   [-1, 10]             5,010
================================================================
Total params: 431,080
Trainable params: 431,080
Non-trainable params: 0
----------------------------------------------------------------
Input size MB: 0.00
Forward/backward pass size MB: 0.12
Params size MB: 1.64
Estimated Total Size MB: 1.76
----------------------------------------------------------------
```

### B.3  PROOF OF WORK

We present the accuracy vs timing of attacks in Figures 7 and 8. We also include similar graphs when using pkNN to compute the privacy cost and these graphs are shown in Figures 9 and 10.

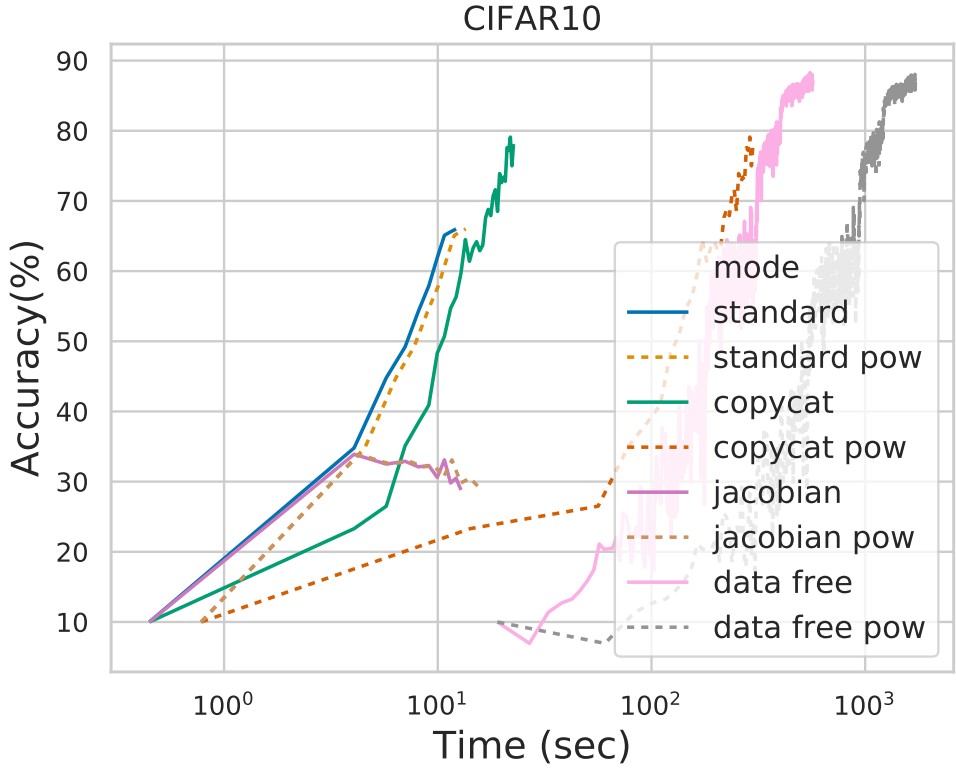

Figure 7: **PoW against model extraction attacks** for the CIFAR10 dataset.

### B.4 Synthetic + In-Distribution Data for Attacks

Apart from attackers who try to minimize their costs to avoid a high computational cost of PoW, we can consider attackers who use a mixture of in-distribution and out-of-distribution data. We call this attack InOut. Here an attacker can first query with out-of-distribution methods and then have intervals where in distribution data (low in quantity) is used to prevent the privacy cost from continually increasing at a much higher rate than a normal user. The usage of both in distribution and out of distribution can also help speed up the extraction process. In Figure 11, we compare this method with CopyCat CNN which randomly queries only using out-of-distribution data and random querying which only uses in distribution data. We observe that the method which targets our attack is able to achieve a lower privacy cost than CopyCat CNN which uses OOD data while also being able to reach a higher accuracy, but is still significantly worse than the random method.

### B.5 PATE privacy cost vs pkNN cost

For the privacy cost, we also consider privacy accounting in the case when there are multiple models behind the ML API which are then used to both make the predictions returned to the model and to compute the privacy. Figure 5 shows the experimental results for multiple models behind the API. We observe that as the number of models in the ensemble increases, the effectiveness of the worstcase method also decreases which is likely because with more models behind the API, the difference in the privacy costs of queries also decreases and therefore an attacker selecting queries with the minimum privacy cost is not able to achieve a better accuracy privacy tradeoff.

In the case of a single model behind an ML API, we can also use a form of PATE proposed by Zhu et al. (2020). First, it computes the representations from training data points obtained in the last (softmax) layer of a neural network. Second, for each new data point to be labeled, its representation is found in the last layer. Finally, in the representation space, we search for the $k$ nearest neighbours from the training set. These $k$ nearest points become the teachers whose votes are their ground-truth

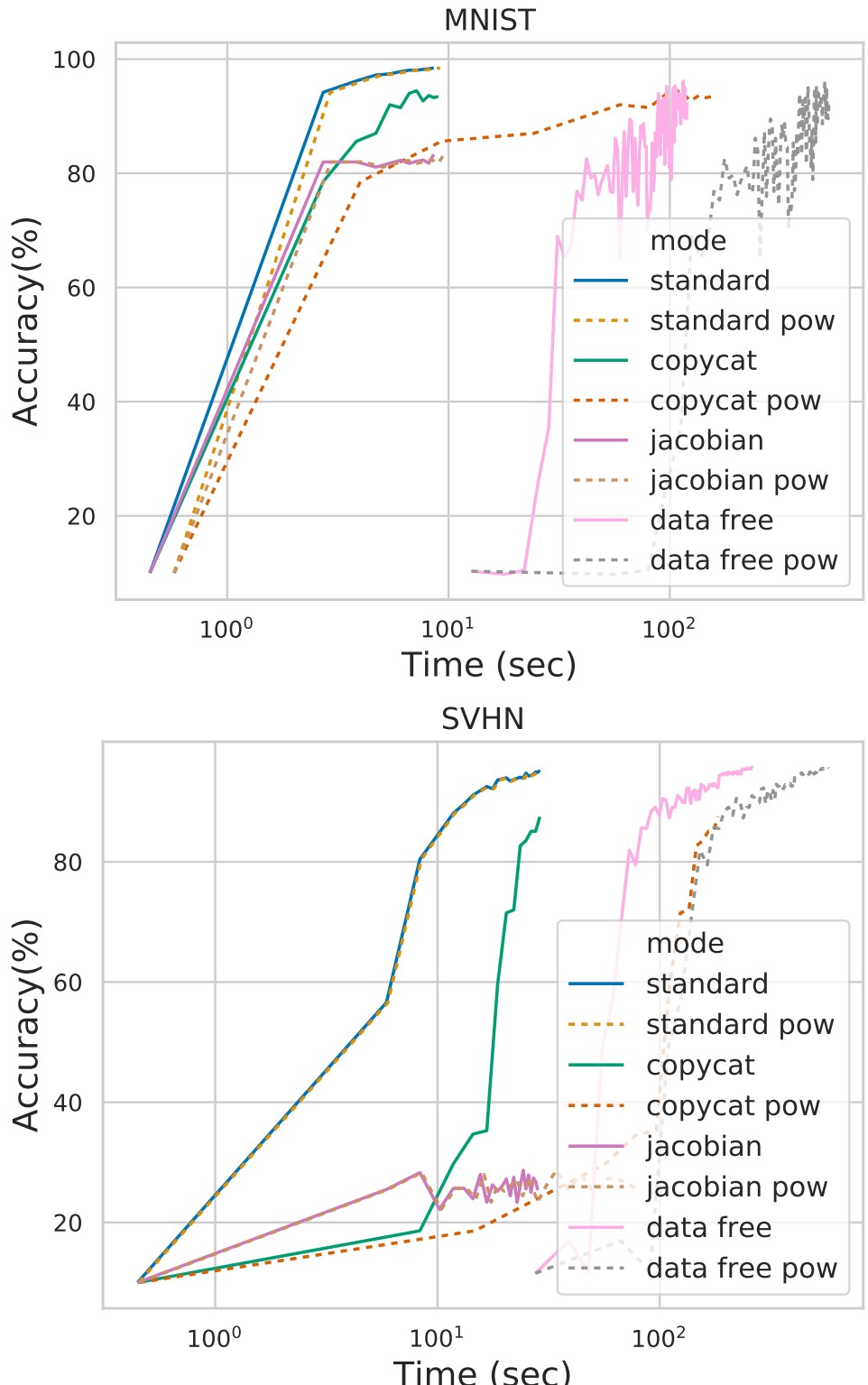

Figure 8: **PoW against model extraction attacks** for MNIST and SVHN datasets.

labels. The remaining steps are the same as in the standard PATE. Similar results are obtained when

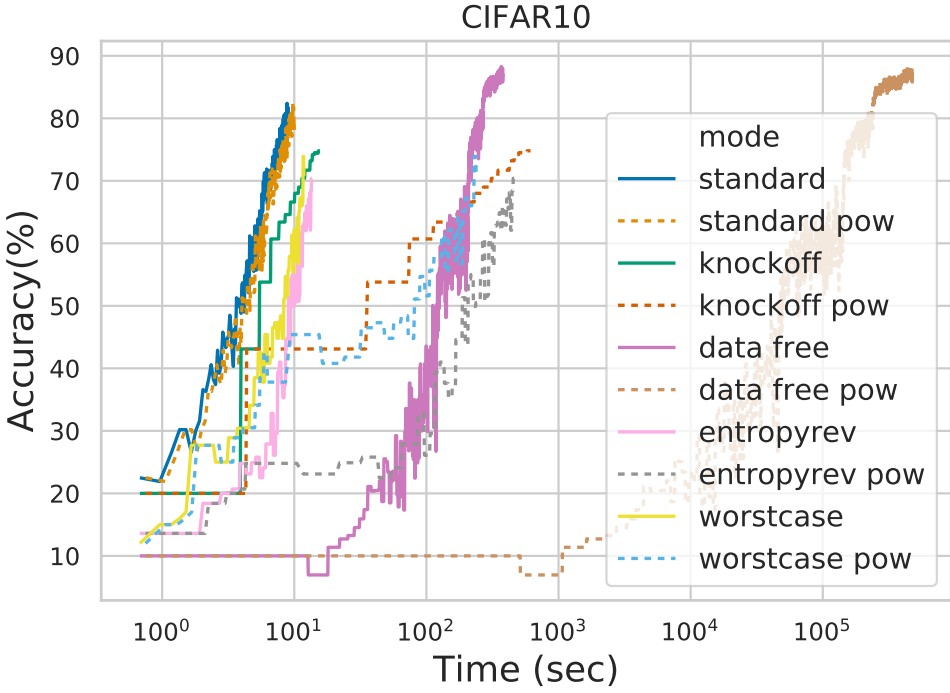

Figure 9: **PoW against model extraction attacks** for the CIFAR10 dataset using the PkNN based privacy cost.

using this method with the main difference being that a larger value of the parameter $a$ is needed as a result of the lower individual privacy costs and thus lower differences obtained with the pkNN cost.

### B.6    OTHER QUERY COST METRICS

For completeness, we also harness metrics from information theory and out-of-distribution detection methods. The entropy and gap metrics give similar estimations of the information leakage and they correspond directly to the entropy and gap methods used for the active learning types of attacks (as described in Section 5). We also use the scores for queries from Generalized ODIN (Hsu et al., 2020), which is one of the state-of-the-art out-of-distribution detectors. Note that using the number of queries only instead of the other metrics is ineffective since after the same number of queries, we observe that attacks such as Knockoff (Orekondy et al., 2019), which query a victim model using out-of-distribution data, can extract model of the same accuracy as a legitimate user querying with in-distribution data. However, Knockoff incurs a much higher privacy cost than a legitimate user. Note that the entropy or gap can be computed using the probability output for a query while the computation of privacy requires access to the whole training set, which makes the privacy metric more secure.

### B.7    ACTIVE LEARNING

This section shows the results of running the active learning methods described in Section 5 on the MNIST, SVHN and CIFAR10 datasets. An iterative training process whereby the attacker model was retrained on new queries is used for the active learning methods whereas for random querying, all queries were done at once after which the model was trained. 50 epochs were used for training MNIST and SVHN models while 200 epochs were used for training CIFAR10 models.

The active learning experiments for the entropy and gap metrics were also replicated on the imagenet dataset (Figure 14) to test the effectiveness of our metrics on larger datasets.

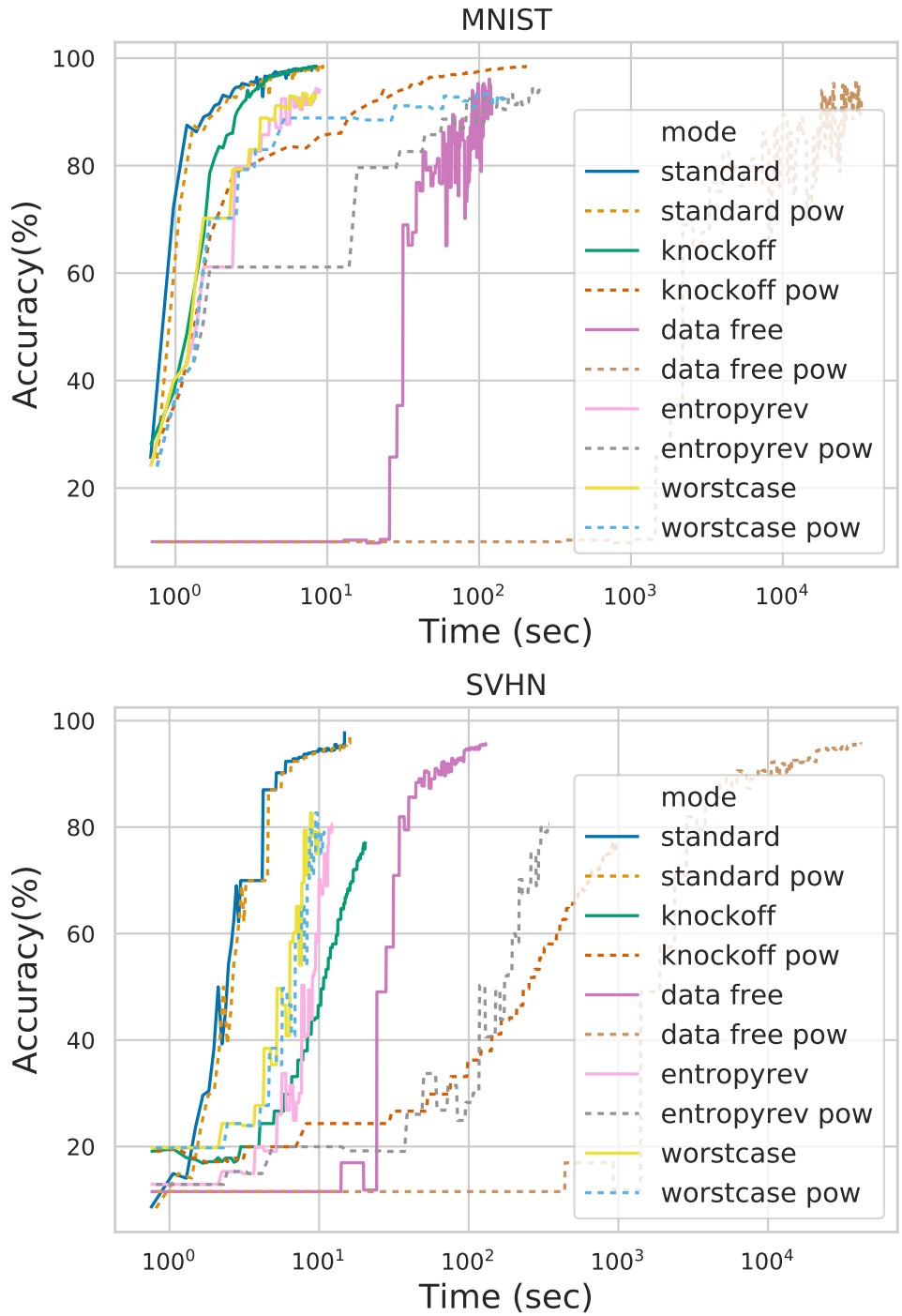

Figure 10: **PoW against model extraction attacks** for MNIST and SVHN datasets using the PkNN based Privacy Cost. Here the worstcase and entropyrev methods use out of distribution data.

### B.8  DATA FREE MODEL EXTRACTION

Figure 15 shows the accuracy of the attacker model as the number of queries increases for the DataFree attack.

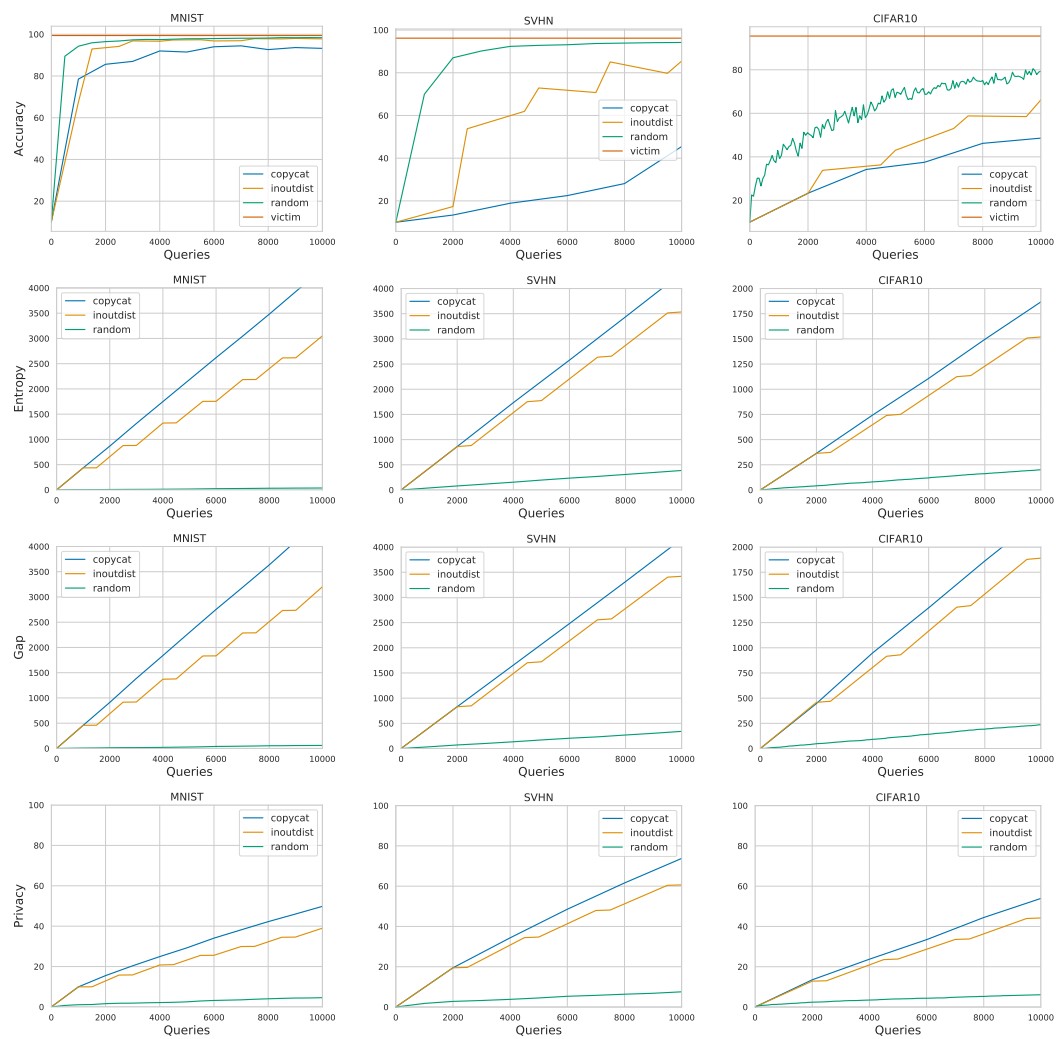

Figure 11: **InOut attack: Accuracy, Entropy, Gap, and Privacy of the victim model on all the answered queries vs number of Queries** for MNIST, SVHN and CIFAR10 datasets.

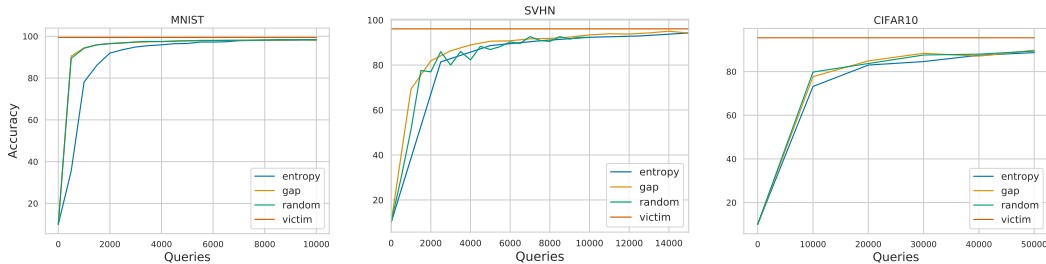

Figure 12: **AL attack: Accuracy of the attacker model vs number of Queries** for MNIST, SVHN and CIFAR10 datasets, respectively. The accuracy of the student model resembles the increase in accuracy for standard training. The higher the model accuracy, the more queries are needed (higher entropy is incurred) to further improve the model.

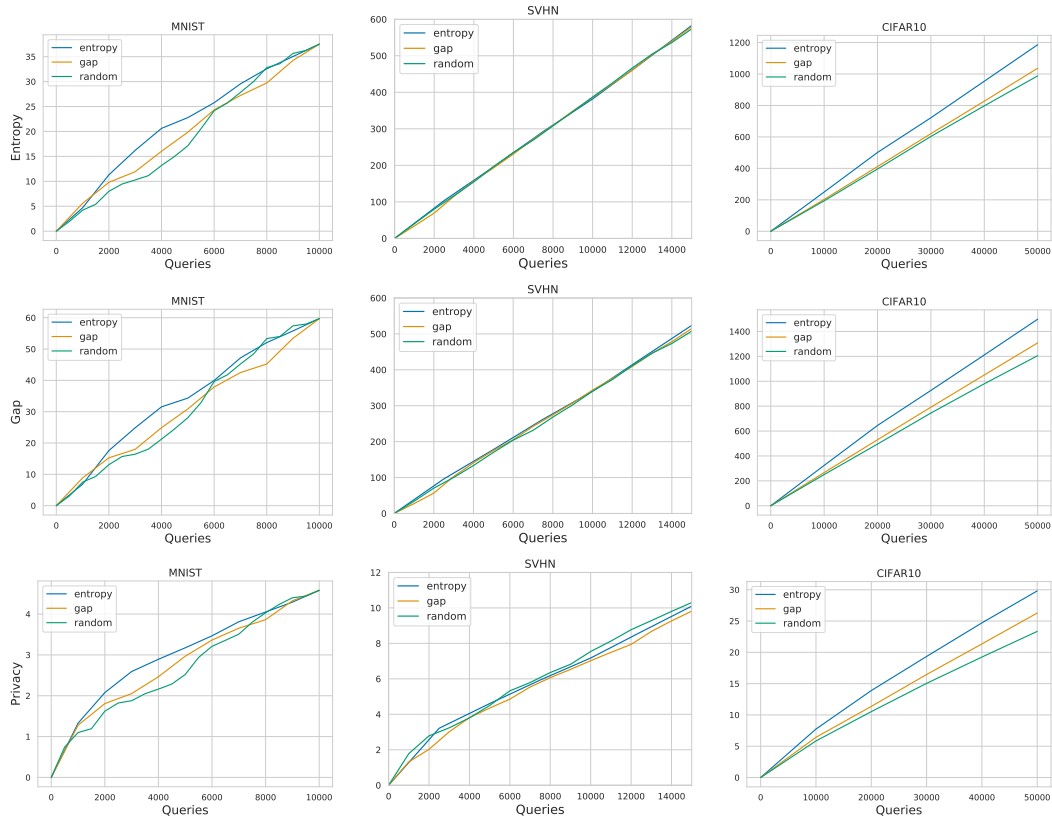

Figure 13: **AL attacks: Entropy, Gap, Privacy costs of the victim model on all the answered queries vs number of Queries** for the MNIST, SVHN and CIFAR10 datasets, respectively. The active learning methods and random querying have similar trends in the privacy metrics. In this case, the gap-based attack helps more to achieve higher accuracy with fewer queries than the entropy attack. The values of the three grow similarly with more queries.

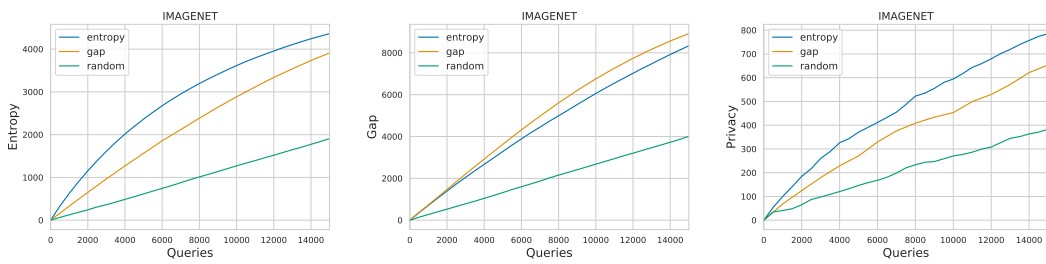

Figure 14: **AL attacks: Entropy, Gap, Privacy costs of the victim model on all the answered queries vs number of Queries** for the Imagenet dataset. It is easy to differentiate between the active learning methods and random querying using all three metrics. Based on these empirical trends, the privacy cost resembles more to entropy than the gap metric.

## B.9   JACOBIAN/ JACOBIAN-TR

Figure 16 shows the accuracy of the attacker model as the number of queries increases for Jacobian and Jacobian-TR attacks. Figure 17 shows the values of the cost metrics against the number of queries.

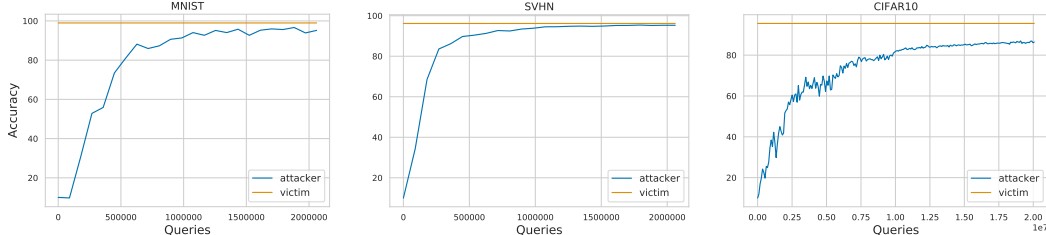

Figure 15: **DataFree attack: Accuracy (of the student/attacker model) vs number of Queries** for MNIST, SVHN, and CIFAR10 datasets, respectively. The accuracy of the student model resembles the increase in accuracy for a standard training. The higher the model accuracy, the more queries are needed (higher entropy is incurred) to further improve the model. DataFree needs 2 mln queries to either match the accuracy of the victim model (for SVHN) or 1.5 mln for MNIST and 20 mln for CIFAR10 to achieve its highest possible accuracy (which is a few percentage points lower than the accuracy of the victim model).

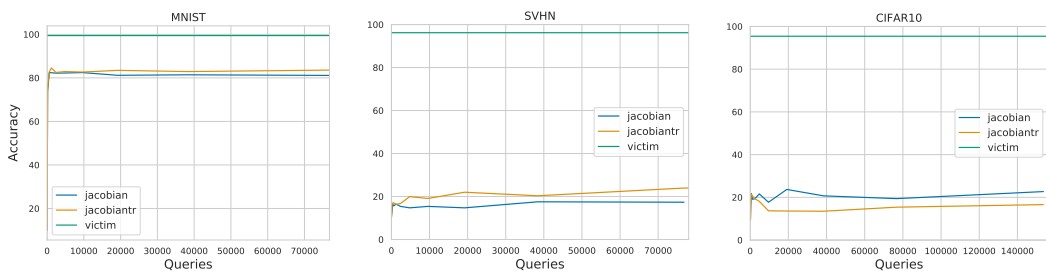

Figure 16: **Jacobian attacks: Accuracy (of the student/attacker model) vs number of Queries** for MNIST, SVHN, and CIFAR10 datasets, respectively. The accuracy of the student model increases by a large amount at the beginning when using small number of initial queries and then remains relatively constant (when there are more Jacobian augmented queries). This is likely because after the initial subset of images, the others are generated based on Jacobian augmentation and thus eventually will not lead to sufficient new information for the model to learn from. Jacobian attack achieves lower accuracy of the extracted model than the other attacks (only about 80% for MNIST and 20% for SVHN as well as CIFAR10).

## B.10    MIXMATCH

Figure 18 shows the accuracy of the attacker model over time when MixMatch is used for training and Figure 19 shows the corresponding values of the cost metrics.

## B.11    KNOCKOFF NETS AND COPYCAT CNN

We note both of these attacks have the same costs as both methods query randomly from a task-specific but an out-of-distribution dataset, in this case ImageNet. Therefore Figure 3 shows the same result for the Knockoff Nets and CopyCat. The accuracy of these methods is compared against other attacks that use out-of-distribution data in Figure 6.

## B.12    FASHION MNIST DATASET

We also run similar experiments on the Fashion-MNIST dataset for which the results are shown in Figures 20, 21 and 22. We note that we have mostly similar trends with the main difference being that the entropy AL method gives a lower cost than random querying. This is likely a result of the attacker's estimation of the costs on the victim side not being as accurate as the other methods which causes the selected queries to not have the smallest cost when measured on the victim's side. In

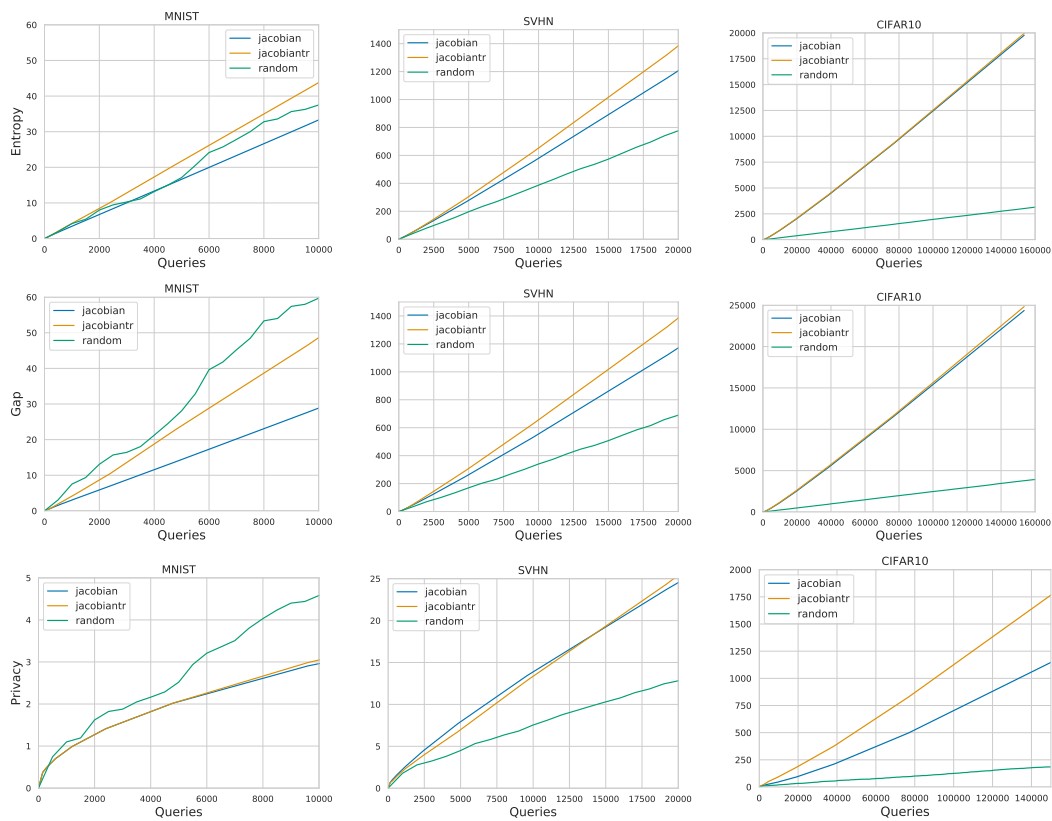

Figure 17: **Jacobian attacks: Entropy, Gap, Privacy costs of the victim model on all the answered queries vs number of Queries** for the MNIST, SVHN and CIFAR10 datasets, respectively. Jacobian methods incur higher costs for SVHN and CIFAR10 while for MNIST, random has a slightly higher cost due to not enough difference of the generated queries from the initial query points.

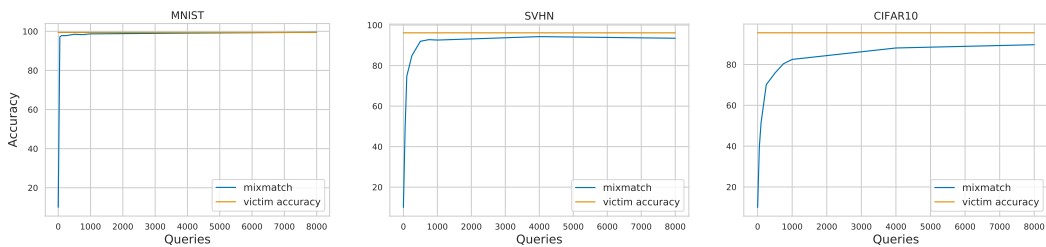

Figure 18: **MixMatch attack: Accuracy (of the student/attacker model) vs number of Queries** for MNIST, SVHN, and CIFAR10 datasets, respectively. A higher number of queries leads to a higher accuracy. The test accuracy corresponding to the epoch with the maximum validation accuracy is used for these plots. MixMatch is the best performing attack in terms of the task accuracy goal. It requires only around a few hundreds queries for MNIST, 1000 for SVHN, and 8000 for CIFAR10, to almost match the accuracy of the victim model.

other words, there is a mismatch between the cost estimated on the attacker's side and the real cost computed on the victim's side.

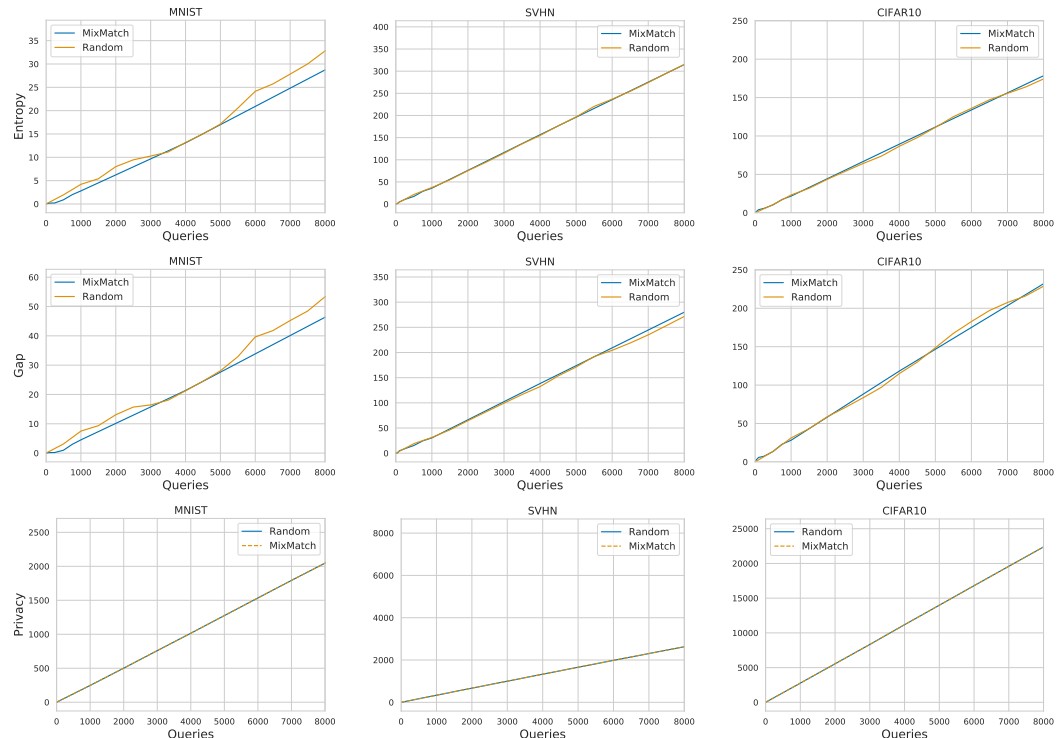

Figure 19: **MixMatch attack: Entropy, Gap, and Privacy costs of the victim model on all the answered queries vs number of Queries** for MNIST, SVHN and CIFAR10 datasets, respectively. The costs are very similar because the MixMatch extraction also uses random querying to create the labeled set.

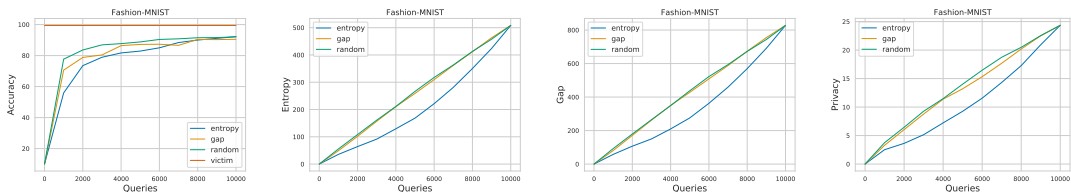

Figure 20: **AL attacks: Accuracy, Entropy, Gap, and Privacy of the victim model on all the answered queries vs number of Queries** for the Fashion-MNIST dataset. Random queries can provide higher accuracy of the extracted model and lower costs incurred by the queries on the victim's side than the simple AL methods.

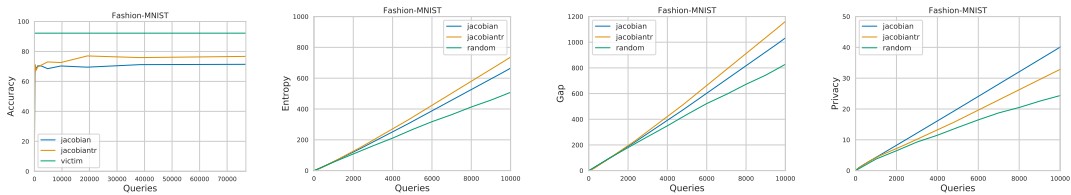

Figure 21: **Jacobian attacks: Accuracy, Entropy, Gap, and Privacy of the victim model on all the answered queries vs number of Queries** for the Fashion-MNIST dataset.

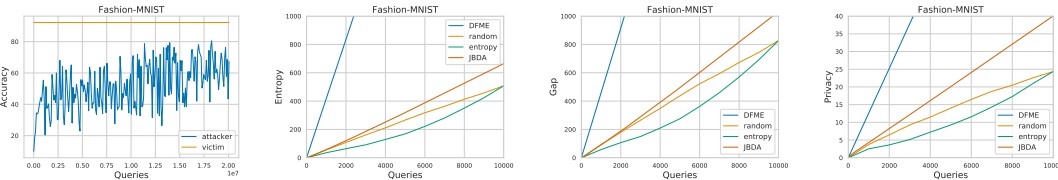

Figure 22: **DataFree Attack: Accuracy, Entropy, Gap, and Privacy of the victim model on all the answered queries vs number of Queries** for the Fashion-MNIST dataset.

Table 5: **Compare computation times (in seconds) of querying the API** when only inference is performed or with the additional computation of the query cost when labeling samples.

| DATASET | # OF LABELED SAMPLES | INFERENCE | PRIVACY | ENTROPY | GAP |
|---|---|---|---|---|---|
| MNIST | 64 | $0.214 \pm 0.000$ | $0.464 \pm 0.004$ | $0.214 \pm 0.001$ | $0.214 \pm 0.001$ |
| SVHN | 64 | $0.250 \pm 0.000$ | $2.167 \pm 0.012$ | $0.250 \pm 0.001$ | $0.250 \pm 0.001$ |
| CIFAR10 | 64 | $0.249 \pm 0.001$ | $1.031 \pm 0.006$ | $0.250 \pm 0.001$ | $0.249 \pm 0.001$ |

## C    COMPUTING THE COST OF QUERIES

We can compute the cost of queries using different measures, such as differential privacy, entropy, or gap. We compare the overheads of each of these methods in terms of their execution time in Table 5.

## D    PRIVACY COST WITH PATE

Differential privacy (DP) is a canonical framework for measuring the privacy leakage of a randomized algorithm (Dwork et al., 2006). It requires the mechanism, the training algorithm in our case, to produce statistically indistinguishable outputs on any pair of adjacent datasets, i.e., datasets differing by only one data point. This bounds the probability of an adversary inferring properties of the training data from the mechanism's outputs.

**Definition 1 (Differential Privacy)** *A randomized mechanism $\mathcal{M}$ with domain $\mathcal{D}$ and range $\mathcal{R}$ satisfies $(\varepsilon, \delta)$-differential privacy if for any subset $\mathcal{S} \subseteq \mathcal{R}$ and any adjacent datasets $d, d' \in \mathcal{D}$, i.e. $\|d - d'\|_1 \leq 1$, the following inequality holds:* $\Pr\left[\mathcal{M}(d) \in \mathcal{S}\right] \leq e^\varepsilon \Pr\left[\mathcal{M}(d') \in \mathcal{S}\right] + \delta$.

PATE (Private Aggregation of Teacher Ensembles) by (Papernot et al., 2018) uses RDP (Rényi Differential Privacy) that generalizes pure DP (i.e., when $\delta = 0$) by using the Réni-divergence to compare the mechanism's output distributions and obtains the required randomization for privacy by sampling from a Gaussian distribution and

**Definition 2** *Rényi Differential Privacy (Mironov, 2017). We say that a mechanism $\mathcal{M}$ is $(\lambda, \varepsilon)$-RDP with order $\lambda \in (1, \infty)$ if for all neighboring datasets $X, X'$:*

$$D_\lambda(\mathcal{M}(X)\|\mathcal{M}(X')) = \frac{1}{\lambda - 1} \log E_{\theta \sim \mathcal{M}(X')}\left[\left(\frac{p_{\mathcal{M}(X)}(\theta)}{p_{\mathcal{M}(X')}(\theta)}\right)^\lambda\right] \leq \varepsilon$$

It is convenient to consider RDP in its functional form as $\varepsilon_{\mathcal{M}}(\lambda)$, which is the RDP $\varepsilon$ of mechanism $\mathcal{M}$ at order $\lambda$. RDP analysis for Gaussian noise is particularly simple.

**Lemma 1** *RDP-Gaussian mechanism. Let $f : \mathcal{X} \to \mathcal{R}$ have bounded $\ell_2$ sensitivity for any two neighboring datasets $X, X'$, i.e., $\|f(X) - f(X')\|_2 \leq \Delta_2$. The Gaussian mechanism $\mathcal{M}(X) = f(X) + \mathcal{N}(0, \sigma^2)$ obeys RDP with $\varepsilon_{\mathcal{M}}(\lambda) = \frac{\lambda \Delta_2^2}{2\sigma^2}$.*

Another notable advantage of RDP over $(\varepsilon, \delta)$-DP is that it composes naturally.

**Lemma 2** *RDP-Composition (Mironov, 2017). Let mechanism $\mathcal{M} = (\mathcal{M}_1, \dots, \mathcal{M}_t)$ where $\mathcal{M}_i$ can potentially depend on outputs of $\mathcal{M}_1, \dots, \mathcal{M}_{i-1}$. $\mathcal{M}$ obeys RDP with $\varepsilon_{\mathcal{M}}(\cdot) = \sum_{i=1}^{t} \varepsilon_{\mathcal{M}_i}(\cdot)$.*

PATE is a model-agnostic approach to DP in ML (Papernot et al., 2017a). PATE trains an ensemble of models on (disjoint) partitions of the training set. Each model, called a teacher, is asked to predict on a query and to vote for one class. Teacher votes are collected into a histogram where $n_i(x)$ indicates the number of teachers who voted for class $i$, $x$ is an input sample. To protect privacy, PATE relies on the noisy argmax mechanism and only reveals a noisy aggregate prediction (rather than revealing each prediction directly): $\mathrm{argmax}\{n_i(x) + \mathcal{N}(0, \sigma_G^2)\}$ where the variance $\sigma_G^2$ of the Gaussian controls the privacy loss of revealing this prediction. Loose data-independent guarantees are obtained through advanced composition (Dwork et al., 2014) and tighter data-dependent guarantees are possible when there is high consensus among teachers on the predicted label (Papernot et al., 2018). To further reduce privacy loss, Confident GNMax only reveals predictions that have high consensus: $\max_i\{n_i(x)\}$ is compared against a noisy threshold also sampled from a Gaussian with variance $\sigma_T^2$. A student model is trained in a semi-supervised fashion on noisy labels returned by the ensemble. Each returned label incurs a privacy loss, which we use in our method to measure the query cost. We train the ensemble of teachers on the defender's training set. We do not reveal the PATE predictions, skip the confident GNMax since the privacy loss has to be computed for each query, and do not train a student model. The predictions are returned using the original defender's model.

# E   MORE INFORMATION ON OTHER DEFENSES

**Active defenses** can be further categorized into *accuracy-preserving*, which do not change the predicted top-1 class, and *accuracy-constrained*, where the drop in accuracy is bound but allows for a higher level of perturbations. Lee et al. (2019) introduce an *accuracy-preserving* defense which perturbs a victim model's final activation layer. The predicted classes are preserved but output probabilities are distorted so that an adversary is forced to use only the labels. *Accuracy-constrained* defenses can also alter the returned label. Orekondy et al. (2020) modify the distribution of model predictions to impede the training process of the stolen copy by poisoning the adversary's gradient signal, however, it requires a costly gradient computation. The adaptive misinformation defense (Kariyappa & Qureshi, 2020) identifies if a query is in-distribution or out-of-distribution (OOD) and then sends incorrect predictions for OOD queries. If an attacker does not have access to in-distribution samples, this degrades the accuracy of the attacker's clone model but also decreases the accuracy for benign users. This defense uses OOD detector trained on both in-distribution and OOD data. Kariyappa et al. (2021) also propose to train an *ensemble* of diverse models that returns correct predictions for in-distribution samples and dissimilar predictions for OOD samples. The defense uses a hash function, which is assumed to be kept secret, to select which model from the ensemble should serve a given query. To train either OOD detector or the ensemble of models, the defenses introduced by Kariyappa et al. assume knowledge of attackers' OOD data that are generally hard to define a-priori and the selection of such data can easily bias learning (Hsu et al., 2020).

**Reactive defenses** are applied after a model has been stolen. Watermarking allows a defender to embed some secret pattern in their model during training (Adi et al., 2018) or inference (Szyller et al., 2019). For the defense to function, any extracted model obtained by the attacker should inherit this secret pattern. However, these watermarks are easy to remove because they are from a different distribution than the training data of the model. Subsequently, Jia et al. (2020a) proposed entangled watermarks to ensure that watermarks are learned jointly with the original training data, which requires changes to the training of the defended model and exhibits a trade-off with model accuracy. Rather than encode additional secret information in the model, two recent lines of work leverage secret information *already* contained in the model. Dataset inference (Maini et al., 2021) identifies whether a given model was stolen by verifying if a suspected adversary's model has private knowledge from the original victim model's dataset. Proof of learning (Jia et al., 2021) involves the defender claiming ownership of a model by showing incremental updates of the model during training process, which is difficult for the attacker to reproduce. Unfortunately, reactive defenses are effective after a model theft and require a model owner to obtain partial or full access to the stolen model. This means that if an attacker aims to use the stolen model as a proxy to attack the victim model, or simply keeps the stolen model secret, it is impossible for the model owner to protect themselves with these defenses.

