# OpenReview forum: "Increasing the Cost of Model Extraction with Calibrated Proof of Work"
_ICLR.cc/2022/Conference — ICLR 2022 Spotlight_

### Official Review · Reviewer_emnb · 2021-10-21

**Correctness:** 3
**Technical Novelty And Significance:** 3
**Empirical Novelty And Significance:** 3
**Recommendation:** 6
**Confidence:** 4

**Main Review:**

This paper proposes a defense against model extraction attacks by forcing users to do a proof-of-work (PoW) puzzle before they receive the labels from the victim model. To minimize the impact on legitimate users, the defense tunes the per-query difficulty based on the extent of information leakage. The privacy cost is evaluated via differential privacy budget. Compared to other defense methods, this method will not reduce the accuracy of answers, and it can protect the model before the model stealing happens (unlike watermark schemes).

My comments are as follows.

Firstly, I think the structure of the paper can be improved. The first two sections of the paper take up a lot of space, so that the discussion in section 3 is not clear enough. Some experimental results are placed in the appendix, and some tables interspersed in the reference list.

Secondly, section 3.3 shows how to use differential privacy to evaluate the privacy cost. Though the authors explain "We compute the privacy cost of queries using the PATE framework",  "The privacy cost is computed based on the consensus among teachers and the amount of privacy noise added to the histogram" and  "more details can be found in appendix D", I would suggest to show the main formulas and conclusions here to help the readers to follow.

Thirdly, the authors create two models. The first one is used to predict query time from the privacy cost, the second one is used to predict the number of leading zeros for the desired time. So the whole mapping flow is : privacy cost → query time → the number of leading zeros. The authors find that using a simple linear regression is sufficient for both models. Firstly, Figure 2 shows the dependency between the computation time and the number of leading zeros is exponential, is it contradictory? Secondly, I think there should be a clear mapping from privacy cost to the number of leading zeros, which will help practitioners easily to tune parameters to control the planned results, such as "Time POW" in Table 2. Thirdly, I can't find the results about these linear regression models in the experiment part.

Finally, the difference in privacy cost between attackers and legitimate users is the main evaluation metric. Figure 3 shows a higher privacy leakage for the attackers compared to a standard user, but for the CIFAR10 dataset, this distinction is not very obvious, and it causes the computational cost for attackers to increase a little bit (In Table 2, line 16, for the attack COPYCAT on CIFAR10, time changes from 148.3 to 326). This situation may be caused by the ambiguous mapping from privacy cost to computational time, and it may pose a risk of defense failure. So more analysis is needed here.


**Summary Of The Paper:**

This paper proposes a novel defense to prevent model stealing by requiring users to solve proof-of-work puzzles. The authors evaluate their method against different types of model extraction attacks. The results show that the defense will result in the attacker costing higher computational time (100x) than legitimate users (2x).


**Summary Of The Review:**

see main review

---

> ### Author Response · Authors · 2021-11-17
> **Mapping from privacy cost to PoW difficulty**
>
> >**I think there should be a clear mapping from privacy cost to the number of leading zeros, which will help practitioners easily tune parameters to control the planned results, such as "Time POW" in Table 2. I can't find the results about these linear regression models in the experiment part.**
>
> In our method, first, we find the total privacy cost incurred and the number of queries issued by a given user. We compare the user’s privacy cost with a legitimate privacy cost for the number of queries. Second, we take the absolute difference $x$ between the privacy costs, and the difference is then used to specify how much time should the user spend on solving a new PoW puzzle. This is done using a simple exponential function $f(x) = a^x$, where $a$ is selected by the model's owner. Finally, we map from the PoW time to the required number of leading zero bits in the puzzle.
>
> A model's owner can collect data about legitimate users. We extrapolate to an arbitrary number of queries to map it to the expected legitimate privacy cost. This is done via a simple linear regression. The model's owner also selects the cost function for PoW. In our case, we use HashCash, for which we compute the expected running time given a number of required zero bits. Because we have to map from an arbitrary PoW time to the number of bits, we interpolate the second dataset by another simple linear regression. This solution is modular and gives model owners a possibility of specifying legitimate users, how much an adversary should be penalized, and the choice of the PoW method.
>
> Note that we do not assume that we know how an attacker might query the victim model. We take the absolute difference which measures a deviation of the current user from the expected behavior. If an attacker is using OOD data, the privacy cost is higher than the standard one. On the other hand, if an attacker is adaptive and tries to minimize the privacy cost, the cost is lower than expected and the attacker is penalized as well.
>
> >**Finally, the difference in privacy cost between attackers and legitimate users is the main evaluation metric. Figure 3 shows a higher privacy leakage for the attackers compared to a standard user, but for the CIFAR10 dataset, this distinction is not very obvious, and it causes the computational cost for attackers to increase a little bit (In Table 2, line 16, for the attack COPYCAT on CIFAR10, time changes from 148.3 to 326). This situation may be caused by the ambiguous mapping from privacy cost to computational time, and it may pose a risk of defense failure. So more analysis is needed here.**
>
> For the CIFAR10 dataset, we are able to create up to 50 PATE teacher models, and for MNIST and SVHN up to 250 teachers (as described in Section 4.2). In general, the more teacher models we can create, the more accurate the estimation of the privacy cost. This causes the observed less precise privacy estimation in the case of CIFAR10 when compared to MNIST or SVHN.
>
> Next, we improved the mapping from the privacy cost to the computational time, which increased the time from 326 to 23921 seconds. We reported the new results in Table 2 and described the mapping in Section 3.3.

---

> ### Author Response · Authors · 2021-11-17
> **Organization of the paper, Privacy cost, Linear models**
>
> Thank you very much for your detailed and insightful feedback. We address individual points below:
>
> >**Firstly, I think the structure of the paper can be improved. The first two sections of the paper take up a lot of space, so that the discussion in section 3 is not clear enough.**
>
> We reduced the size of the first two Sections (Introduction and especially, the Related Work) and expanded Section 3 to make it more clear. If the reviewer has any specific points they think need more detail, we would be happy to further revise the manuscript.
>
> >**Some experimental results are placed in the appendix, and some tables are interspersed in the reference list.**
>
> We separated the reference list and the results from the appendix so that they do not overlap.
>
> Regarding the experimental results, we include more graphs in the Appendix for completeness, and for clarity so that they are formatted large enough to be readable. If the reviewer has further suggestions on which experimental results should be moved back into the main paper, we will decrease the size of other sections further to make room.
>
> >**Secondly, section 3.3 shows how to use differential privacy to evaluate the privacy cost. Though the authors explain "We compute the privacy cost of queries using the PATE framework", "The privacy cost is computed based on the consensus among teachers and the amount of privacy noise added to the histogram" and "more details can be found in Appendix D", I would suggest showing the main formulas and conclusions here to help the readers to follow.**
>
> We would like to thank the reviewer for this remark. In our revised manuscript, we added more details directly in Section 3.3, specifically:
> “PATE creates teacher models using the defender's training data and predictions from each teacher are aggregated into a histogram where $n_i$ indicates the number of teachers who voted for class $i$. The privacy cost is computed based on the consensus among teachers and it is smaller when more teachers agree. The consensus is high if we have a low probability that the label with the maximum number of votes is not returned. This probability is expressed as: $\frac{1}{2} \sum_{i \ne i^*} \text{erfc}(\frac{n_{i^*} - n_{i}}{2\sigma})$, where $i^*$ is an index of the correct class, $\text{erfc}$ is the complementary error function, and $\sigma$ is the privacy noise (we explain the details in Appendix D). Then a student model is trained by transferring knowledge acquired by the teachers in a privacy-preserving way. This student model corresponds to the attacker’s model trained during extraction. [Papernot et al. (2017a)](https://arxiv.org/pdf/1610.05755.pdf) present the analysis (e.g., Table 2) of how the performance (in terms of accuracy) of the student model increases when a higher privacy loss is incurred through the answered queries.”
>
> | Dataset | $\varepsilon$ | Student’s accuracy |
> |---------|---------------|--------------------|
> | MNIST   | 2.04          | 98.00%             |
> | MNIST   | 8.03          | 98.10%             |
> | SVHN    | 5.04          | 82.72%             |
> | SVHN    | 8.19          | 90.66%             |
>
> If the reviewer thinks we should add other formulas to Section 3.3, we would be happy to further revise our manuscript.
>
> >**The authors find that using a simple linear regression is sufficient for both models. Firstly, Figure 2 shows the dependency between the computation time and the number of leading zeros is exponential, is it contradictory?**
>
> We thank the reviewer for this question and have revised the manuscript in section 3.2 to make this more clear. We take the logarithm of the computation time, which transforms this exponential dependency between the computation time and the number of leading zeros to a linear dependency. In our case, modeling this dependency with a linear regression is sufficient.

---

> ### Author Response · Authors · 2021-11-23
> **Have concerns been addressed?**
>
> We would like to follow up on our answers, especially regarding the Proof-of-Work (PoW) difficulty, privacy cost, mapping from privacy to PoW, and the organization of the paper. Do our replies adequately address the reviewer's concerns?

---

### Official Review · Reviewer_LLkZ · 2021-10-28

**Correctness:** 3
**Technical Novelty And Significance:** 3
**Empirical Novelty And Significance:** 2
**Recommendation:** 3
**Confidence:** 4

**Main Review:**

Strengths
- S1: This paper proposes a novel defense focusing on delaying the model stealing process.
- S2: The proposed defense is tested against 6 attacks on 3 datasets.

Weaknesses
- W1: If the goal of the proof of work is just to slow down the process, the server can simply choose to delay the response, unlike the distributed block chain. Thus, the proof of work does not look necessary.
- W2: The defense does not work against an attacker using in-distribution data, such as a subscribed user using the model normally until enough data is accumulated.
- W3: The use of PATE as the information leakage estimator is not justified. PATE is devised to protect sensitive data, and how sensitive data is connected to the performance of a machine learning model is unclear. This component is rather the most important part of the framework as it dictates the user experience and the attacker's success. A detailed reasoning or comparison to other options should improve the paper.

**Summary Of The Paper:**

This paper proposes a new defense to the model stealing problem that an attacker queries a supervised machine learning model service to have data labeled to use as training data and copy the functionality of the model. The proposed method is proactive defense that slows down the attacker in obtaining the labeled data, based on the information leakage estimator. The evaluation shows that existing attacks using out-of-distribution data can be slow down more than a regular user querying in-distribution data.

**Summary Of The Review:**

This paper is in general well written, and the experiment was done to show the strengths and the weaknesses of the proposed defense. The defense against model stealing is inherently difficult problem where the attacker in the end can steal the model using the queried data and returned label pairs unless architecture or resource is no object (e.g., if not GPT-3). The main idea proposed in the paper against such a strong attack looks interesting, but the key technique used here is the proof of work. I am not convinced this is necessary at all since the original model is hosted and controlled centrally, and response time can be fully controlled without using the proof of work. Just delaying the response would have much better control of information leakage per time unit than the proof of work that a powerful machine can do faster, or simultaneously processing multiple queries with multiple machines reaching up to the speed of a regular user. Rather the important piece is the information leakage estimator which is somewhat overlooked in the paper, especially in terms of why PATE has to be the right choice here. I think focusing on this can lead to a better paper.

---

### Official Review · Reviewer_hR7i · 2021-11-02

**Correctness:** 4
**Technical Novelty And Significance:** 3
**Empirical Novelty And Significance:** 2
**Recommendation:** 8
**Confidence:** 4

**Main Review:**

Strengths:
- Interesting idea of applying proof-of-work to machine learning as an entry barrier.
- The novelty of the paper seems to reside in the application of existing concepts, like PATE for privacy cost estimation and HashCash as challenge, as components of a theft defense method. I find this novel enough and quite ingenious. Moreover, the existing concepts were appropriately adapted for the task at hand.
- The experimental section is extensive, and the experimental protocol seems appropriate.
- The paper is clear, well organized and pleasant to read.
- Good quality code base allowing to reproduce the experiments.

Weaknesses:
- The proposed defense does not seem effective against attackers that have access to data from a similar distribution to the training one (e.g., MixMatch baseline).
- The Knockoff attack does not seem to use its most effective querying strategy (Random is used instead of Adaptive).
- Doubling computation cost for a legitimate user is not a prohibitive cost, but is also non-negligible, both from an effort and computation time perspective, as well as the CO2 emissions mentioned in the ethics statement of the paper.

Other comments / questions:
- With the proposed approach, does it mean that legitimate users would incur higher query cost when submitting many queries?
- The paper states multiple times that it proposes the first defense to leave model accuracy intact, but I would argue that it is the first *active* defense to do so.

[Update post-discussion] I am raising my rating following the exchanges below.

**Summary Of The Paper:**

This paper proposes a novel defense against model extraction attacks. The proposed approach slows information leakage by asking all users to answer a puzzle before receiving the response to their query (proof-of-work). The difficulty of the puzzle allows to keep the computation overhead low for legitimate users, while rendering information leakage prohibitively expensive for attackers. The puzzle difficulty is calibrated based on an estimate of how much information each user has already acquired. This is based on PATE, a differential privacy metric. Experiments are performed on multiple datasets, opposing the proposed defense to a wide range of attacks, including adaptive adversaries.

**Summary Of The Review:**

Novel active strategy to defend against model extraction, good experimental results.

---

> ### Author Response · Authors · 2021-11-15
> **The Knockoff attack**
>
> We thank the reviewer for the insightful comments. We address individual points below one by one:
>
> >_**The Knockoff attack does not seem to use its most effective querying strategy (Random is used instead of Adaptive).**_
>
> Thank you for the comment, we ran additional experiments following your suggestion. For context, we initially observed that the adaptive Knockoff attack shows only marginal improvements with ImageNet in the case of Caltech256, CUBS200, and Indoor67 datasets and even performs worse for Diabetic5, as shown in [the original paper (Table 2, Open ILSVRC)](https://bit.ly/3koxLLn). In addition, the code for the adaptive attack was not released in the original implementation and one implementation of it we did find was not scalable for the large number of queries we operate within our testing. Specifically, we used the adversarial-robustness-toolbox library: https://github.com/Trusted-AI/adversarial-robustness-toolbox (ART) to test the adaptive method and we faced out-of-memory errors close to 10000 queries. We can try to investigate it further but think this is a result of the requirement in the RL aspect of the algorithm to retrain the stolen model for every single query sent to the victim model. Thus, we did not include an evaluation of the adaptive method in our original submission.
>
> That said, following the reviewer’s comment, we did compare the random and adaptive methods for a smaller number of queries on the MNIST dataset (using CIFAR10 and SVHN as the attacker’s datasets) and with the CIFAR10 dataset using CIFAR100 dataset as the attacker’s dataset. In all cases, we observed a better attacker performance with the random method (see the results below). Furthermore, the adaptive method assumes access to the ground truth labels for the dataset used for querying. While this may be possible if a common dataset e.g. Imagenet is used, this is not necessarily true for all datasets an attacker can use. Therefore, the adaptive attack (from the ADT library) not only performs worse than random but also requires more adversarial knowledge compared to all other attacks which assume no access to the true labels for the attacker dataset used.
>
> MNIST victim model accuracy: 98.5 %. Using CIFAR10 as the OOD dataset:
>
> |Number of Queries|Accuracy of Stolen Model (Random)|Accuracy of Stolen Model (Adaptive)|
> |-----------------|---------------------------------|-----------------------------------|
> |2000             |64.70%                           |54.50%                             |
> |4000             |67.90%                           |62.20%                             |
> |6000             |58.60%                           |43.80%                             |
> |8000             |78.50%                           |70.10%                             |
>
> MNIST victim model accuracy: 98.5 %. Using SVHN as the OOD dataset:
>
> |Number of Queries|Accuracy of Stolen Model (Random)|Accuracy of Stolen Model (Adaptive)|
> |-----------------|---------------------------------|-----------------------------------|
> |2000             |24.2 %                           |8.9 %                              |
> |4000             |36.40%                           |25.80%                             |
> |6000             |53 %                             |13.2 %                             |
> |7000             |45.2 %                           |25.80%                             |
>
> CIFAR10 Victim model accuracy: 47.6 % (the initial result for a low victim's accuracy). Using CIFAR100 as the OOD dataset:
>
> |Number of Queries|Accuracy of Stolen Model (Random)|Accuracy of Stolen Model (Adaptive)|
> |-----------------|---------------------------------|-----------------------------------|
> |1000             |29.9 %                           |25.70%                             |
> |2000             |34.7 %                           |24.1 %                             |

---

> > ### Author Response · Authors · 2021-11-15
> > **MixMatch, Computation Time & Query Cost**
> >
> > >_**The proposed defense does not seem effective against attackers that have access to data from a similar distribution to the training one (e.g., MixMatch baseline).**_
> >
> > In the case where an attacker has access to in distribution data and acts in the same way as a legitimate user, preventing model extraction is not possible. This is because we are not able to differentiate the queries from a standard user vs an attacker. Unless we negatively impact the accuracy of labels returned to standard users, the attacker in such a case cannot be defended against. Nevertheless, we evaluated the in-distribution adversary to obtain a worst-case evaluation of our approach.
> >
> > >**Doubling computation cost for a legitimate user is not a prohibitive cost, but is also non-negligible, both from an effort and computation time perspective, as well as the CO$_2$ emissions mentioned in the ethics statement of the paper.**
> >
> > To address the problem, our approach can be adapted to rely on PoET (Proof-of-Elapsed-Time) instead of PoW. The users’ resources (e.g., a CPU) would have to be occupied for a specific amount of time without executing any work. At the end of the waiting time, the user sends proof of the elapsed time instead of proof of work. Both proofs can be easily verified. [PoET](https://bit.ly/3bS7Qat) reduces the potential energy wastage in comparison with PoW but requires access to specialized hardware, namely new secure CPU instructions that are becoming widely available in consumer and enterprise processors. If a user does not have such hardware on-premise, the proof of elapsed time could be produced using a service exposed by a cloud provider (e.g., [Azure VMs that feature TEE](https://bit.ly/3D3YBzG)).
> >
> > As far as the overhead factor of 2 is concerned, previous work [[1](https://openreview.net/forum?id=SyevYxHtDB),[2](https://bit.ly/3wG0YX6)] and our results suggest that there is *no free lunch*: the active defenses against model extraction either sacrifice the accuracy of the returned answers or force users to do more work. The principal strength of our active defense approach is that it does not sacrifice accuracy, which thus comes at the expense of computational overhead.
> >
> > We also added the above answer to the Ethics Statement.
> >
> > Furthermore, we decreased the cost for legitimate users in Table 2 from 2x to 1.23x and also tested that even with more queries their cost does not exceed 2x.
> >
> > [1] [Prediction Poisoning: Towards Defenses Against DNN Model Stealing Attacks.](https://openreview.net/forum?id=SyevYxHtDB) Tribhuvanesh Orekondy, Bernt Schiele, Mario Fritz. ICLR 2020.
> >
> > [2] [Defending Against Model Stealing Attacks With Adaptive Misinformation](https://bit.ly/3wG0YX6). Sanjay Kariyappa, Moinuddin K. Qureshi. CVPR 2020.
> >
> > >**With the proposed approach, does it mean that legitimate users would incur higher query costs when submitting many queries?**
> >
> > We modified the approach to make sure that the legitimate user does not incur a higher query cost when submitting many queries. The general idea is that we treat the privacy cost of a legitimate user for a given number of queries as a baseline and the more a given user deviates from this baseline, the higher query cost is incurred. We provide a more detailed description of our approach in Section 3.3.
> >
> > We do not make it impossible to extract a model, but try to create a defense so that no advantage can be gained by selecting queries in a specific way. The defense is successful when an attacker is mapped back to the standard information gain that a legitimate user has. For an adversarial user, the required PoW should be higher to dissuade such a user from an attack (or incentivize the user to act legitimately). In the standard PATE framework, we train a student model, which we can think of as an attacker’s model. The teacher models return predictions to queries that are then used to train the student. After reaching a specified privacy threshold $\varepsilon$, teachers stop answering the queries. In our defense, we measure the privacy leakage incurred by a given user. For a legitimate user, there is also a threshold $\varepsilon$ after which, the user could easily train its own model. The provider of the ML model can define what is expected from a legitimate user by adjusting the $\varepsilon$ value.
> >
> > >**The paper states multiple times that it proposes the first defense to leave model accuracy intact, but I would argue that it is the first active defense to do so.**
> >
> > Thank you for pointing this out. We amended the paper to clarify that in the class of active methods, this is the first approach that leaves the model accuracy intact.

---

> > > ### Author Response · Authors · 2021-11-23
> > > **Have concerns been addressed?**
> > >
> > > We also wanted to ask: do our replies adequately address the reviewer's concerns regarding MixMatch, Computation Time, and Query Cost?

---

> > > > ### Comment · Reviewer_hR7i · 2021-11-23
> > > > **Thank you for all your answers**
> > > >
> > > > I appreciate the authors answering in detail the reviewers' questions. I think addressing these in the paper has added value. My concerns have been answered, I am thus raising my rating by one point.

---

> > > > > ### Author Response · Authors · 2021-11-23
> > > > > **Thank you for the review & increasing the score**
> > > > >
> > > > > We thank the reviewer for the discussion and appreciate the positive feedback.

---

> > ### Author Response · Authors · 2021-11-23
> > **Full results for the Knockoff Adaptive attack**
> >
> > We solved the problem with the out-of-memory error by modifying [the ART library](https://github.com/Trusted-AI/adversarial-robustness-toolbox/blob/main/art/attacks/extraction/knockoff_nets.py). We replaced the explicit collection of sampled data points with their indices. This additional level of indirection helped us to save a substantial amount of memory. We updated Figure 4 and Table 2 in the manuscript with the additional results. Overall, we find that the Knockoff Random attack still works better than Knockoff Adaptive.

---

> > > ### Comment · Reviewer_hR7i · 2021-11-23
> > > **Thank you for the additional experimental results**
> > >
> > > I appreciate the inclusion of results for Knockoff Adaptive attack here and in the paper, especially in view of the effort that seems to have gone into adapting the ART implementation. My comment on this topic has fully been addressed.

---

> > > > ### Author Response · Authors · 2021-11-23
> > > > **Additional Experimental Results: the Knockoff Adaptive attack**
> > > >
> > > > Thank you for the confirmation, we appreciate it.

---

### Official Review · Reviewer_aHDe · 2021-11-05

**Correctness:** 3
**Technical Novelty And Significance:** 3
**Empirical Novelty And Significance:** 3
**Recommendation:** 8
**Confidence:** 3

**Main Review:**

The paper discussed the issue, and the background works well. The purpose is clear, and
there are a satisfying amount of experiments for validating the proposed solution. However,
the privacy scoring section is not clear. I cannot find out why PATE is the best option for
calculating privacy metrics and the exact functionality of this section is not as straightforward
as other parts. Moreover, identification of users is essential since, as mentioned, the difficulty of
POW will increase gradually, but this concern is not well pointed in the paper.

**Summary Of The Paper:**

The paper discusses ML model extraction attacks while using APIs for accessing them on the
public networks. Although some methods exist for preventing or making the attacks hard, all of
them have substantial impacts on legitimate users’ experience while using the system, including
the slower models or lower accuracy in results. This paper proposed a method for dissuading
the attacker by increasing the cost of the attack. Solving a puzzle for all users before getting
the final response (POW) would be the solution noting that the difficulty will be increased if
the system identifies any adverse behaviors. This method requires no modification of the victim
model and can be applied by machine learning practitioners to guard their publicly exposed
models against being easily stolen.

**Summary Of The Review:**

The authors should make the privacy scoring module more clear and add more discussion about identifying malicious nodes in continuous requests

---

> ### Author Response · Authors · 2021-11-15
> **PATE, privacy scoring, and identification of users**
>
> We appreciate the positive, encouraging, and constructive feedback. We thank the reviewer for the detailed analysis of our paper and below provide a case-by-case response to the comments.
>
> >_**I cannot find out why PATE is the best option for calculating privacy metrics.**_
>
> We have modified the manuscript and added this statement in Section 3.3:
>
> There are two canonical approaches to obtaining privacy in deep learning: DP-SGD and PATE. DP-SGD is not applicable in our case as it is an algorithm-level mechanism used during training, which does not measure privacy leakage for individual test queries. On the other hand, PATE is an output level mechanism, which allows us to measure per-query privacy leakage.
>
> >_**The authors should make the privacy scoring module more clear. The exact functionality of this section is not as straightforward as other parts.**_
>
> The goal of the privacy scoring module is to explain how to compute the privacy per query and the way the difficulty of a PoW puzzle is selected. We further extended the explanation of the privacy scoring in Section 3.3:
>
> PATE creates teacher models using the defender's training data and predictions from each teacher are aggregated into a histogram where $n_i$ indicates the number of teachers who voted for class $i$. The privacy cost is computed based on the consensus among teachers and it is smaller when more teachers agree. The consensus is high if we have a low probability that the label with the maximum number of votes is not returned. This probability is expressed as: $\frac{1}{2} \sum_{i \ne i^*} \text{erfc}(\frac{n_{i^*} - n_{i}}{2\sigma})$, where $i^*$ is an index of the correct class, $\text{erfc}$ is the complementary error function, and $\sigma$ is the privacy noise (we explain the details in Appendix D). Then a student model is trained by transferring knowledge acquired by the teachers in a privacy-preserving way. This student model corresponds to the attacker’s model trained during extraction. [Papernot et al. (2017a)](https://arxiv.org/pdf/1610.05755.pdf) present the analysis (e.g., Table 2) of how the performance (in terms of accuracy) of the student model increases when a higher privacy loss is incurred through the answered queries.
>
> | Dataset | $\varepsilon$ | Student’s accuracy |
> |---------|---------------|--------------------|
> | MNIST   | 2.04          | 98.00%             |
> | MNIST   | 8.03          | 98.10%             |
> | SVHN    | 5.04          | 82.72%             |
> | SVHN    | 8.19          | 90.66%             |
>
> If the reviewer has any other specific suggestions, we would be happy to further revise the manuscript.
>
> >_**The identification of users is essential since, as mentioned, the difficulty of PoW will increase gradually, but this concern is not well pointed in the paper.**_
>
> Thank you for pointing this out. We modified the manuscript and added into Section 3.2 the following statement: *Since our defense works on a per-user basis, it requires the identification of users.*
>
> At the end of Section 4.4, we also state: "Another adaptive adversary can attack our operation model with multiple fake accounts instead of funneling all their queries through a single account or there can be multiple colluding users. Our approach not only defends against this because the attacker still needs to solve the proof-of-work for each of the user accounts, but it also simplifies the task of identifying colluding users. We note that one appealing aspect of our defense is that it relies on the privacy cost that can easily be summed across users because of the composition property of differential privacy.”

---

> > ### Author Response · Authors · 2021-11-23
> > **Have concerns been addressed?**
> >
> > We would like to follow up on our responses, especially on the privacy cost and the identification of users. Do they adequately address the reviewer's concerns?

---

### Author Response · Authors · 2021-11-25
**Pending questions**

We would like to thank the reviewers for their questions and comments. The paper has definitely improved as a result. We would like to check one last time if there are any pending questions that we have not adequately addressed.

---

### Decision · Program_Chairs · 2022-01-20

**Decision:**

Accept (Spotlight)

**Comment:**

the paper proposed a novel idea of  requiring users to complete a proof-of-work before they can read the model's prediction to prevent model extraction attacks. Reviewers were excited about the paper and ideas. Some misunderstanding raised by reviewers were sufficiently clarified by the authors in the rebuttal.